# 🪆 Matryoshka Diffusion Models

**Jiatao Gu, Shuangfei Zhai, Yizhe Zhang, Josh Susskind & Navdeep Jaitly**
Apple
{jgu32,szhai,yizzhang,jsusskind,njaitly}@apple.com

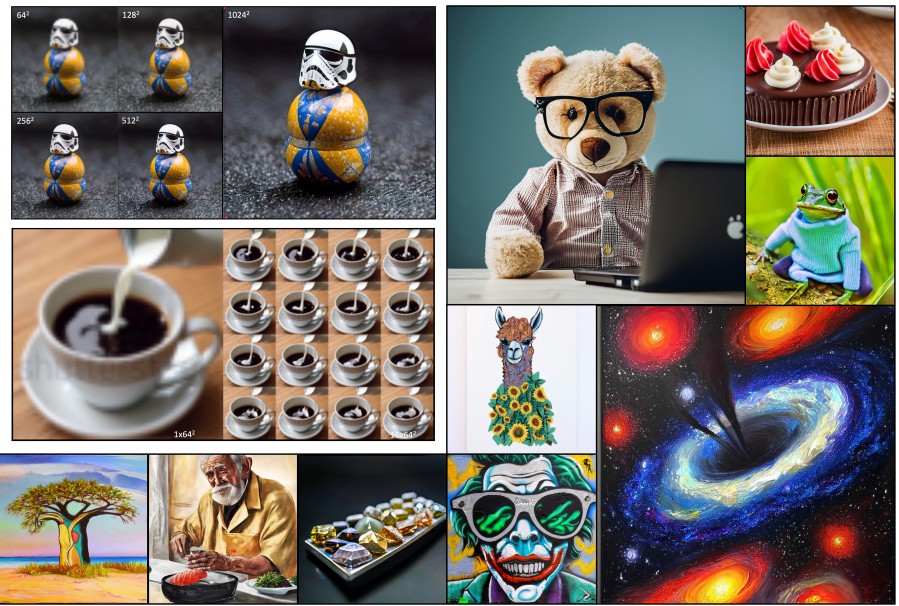

Figure 1: (←↑) Images generated by MDM at $64^2$, $128^2$, $256^2$, $512^2$ and $1024^2$ resolutions using the prompt *"a Stormtrooper Matryoshka doll, super details, extreme realistic, 8k"*; (←↓) 1 and 16 frames of $64^2$ video generated by our method using the prompt *"pouring milk into black coffee"*; All other samples are at $1024^2$ given various prompts. Images were resized for ease of visualization.

## Abstract

Diffusion models are the *de-facto* approach for generating high-quality images and videos but learning high-dimensional models remains a formidable task due to computational and optimization challenges. Existing methods often resort to training cascaded models in pixel space, or using a downsampled latent space of a separately trained auto-encoder. In this paper, we introduce Matryoshka Diffusion (MDM), a novel framework for high-resolution image and video synthesis. We propose a diffusion process that denoises inputs at multiple resolutions jointly and uses a NestedUNet architecture where features and parameters for small scale inputs are nested within those of the large scales. In addition, MDM enables a progressive training schedule from lower to higher resolutions which leads to significant improvements in optimization for high-resolution generation. We demonstrate the effectiveness of our approach on various benchmarks, including class-conditioned image generation, high-resolution text-to-image, and text-to-video applications. Remarkably, we can train a *single pixel-space model* at resolutions of up to $1024 \times 1024$ pixels, demonstrating strong zero shot generalization using the CC12M dataset, which contains only 12 million images. Code and pre-trained checkpoints are released at https://github.com/apple/ml-mdm.

## 1 Introduction

Diffusion models (Sohl-Dickstein et al., 2015; Ho et al., 2020; Nichol & Dhariwal, 2021; Song et al., 2020) have become increasingly popular tools for generative applications, such as image (Dhariwal

& Nichol, 2021; Rombach et al., 2022; Ramesh et al., 2022; Saharia et al., 2022), video (Ho et al., 2022c;a), 3D (Poole et al., 2022; Gu et al., 2023; Liu et al., 2023b; Chen et al., 2023), audio (Liu et al., 2023a), and text (Li et al., 2022; Zhang et al., 2023) generation. However scaling them to high-resolution still presents significant challenges as the model must re-encode the entire high-resolution input for each step (Kadkhodaie et al., 2022). Tackling these challenges necessitates the use of deep architectures with attention blocks which makes optimization harder and uses more resources.

Recent works (Jabri et al., 2022; Hoogeboom et al., 2023) have focused on efficient network architectures for high-resolution images. However, none of the existing methods have shown competitive results beyond $512 \times 512$, and their quality still falls behind the main-stream cascaded/latent based methods. For example, DALL-E 2 (Ramesh et al., 2022), IMAGEN (Saharia et al., 2022) and eDiff-I (Balaji et al., 2022) save computation by learning a low-resolution model together with multiple super-resolution diffusion models, where each component is trained separately. On the other hand, latent diffusion methods (LDMs) (Rombach et al., 2022; Peebles & Xie, 2022; Xue et al., 2023) only learn low-resolution diffusion models, while they rely on a separately trained high-resolution autoencoder (Oord et al., 2017; Esser et al., 2021). In both cases, the multi-stage pipeline complicates training & inference, often requiring careful tuning of hyperparameters.

In this paper, we present Matryoshka Diffusion Models (MDM), a novel family of diffusion models for high-resolution synthesis. Our main insight is to include the low-resolution diffusion process as part of the high-resolution generation, taking similar inspiration from multi-scale learning in GANs (Karras et al., 2017; Chan et al., 2021; Kang et al., 2023). We accomplish this by performing a joint diffusion process over multiple resolution using a Nested UNet architecture ( (see Fig. 2 and Fig. 3). Our key finding is that MDM, together with the Nested UNets architecture, enables 1) a multi-resolution loss that greatly improves the speed of convergence of high-resolution input denoising and 2) an efficient progressive training schedule, that starts by training a low-resolution diffusion model and gradually adds high-resolution inputs and outputs following a schedule. Empirically, we found that the multi-resolution loss together with progressive training allows one to find an excellent balance between the training cost and the model's quality.

We evaluate MDM on class conditional image generation, and text conditioned image and video generation. MDM allows us to train high-resolution models without resorting to cascaded or latent diffusion. Ablation studies show that both multi-resolution loss and progressive training greatly boost training efficiency and quality. In addition, MDM yield high performance text-to-image generative models with up to $1024^2$ resolution, trained on the reasonably small CC12M dataset. Lastly, MDM generalize gracefully to video generation, suggesting generality of our approach.

## 2 DIFFUSION MODELS

**Diffusion models (Sohl-Dickstein et al., 2015; Ho et al., 2020)** are latent variable models given a pre-defined posterior distribution (named the forward diffusion process), and trained with a denoising objective. More specifically, given a data point $\boldsymbol{x} \in \mathbb{R}^N$ and a fixed signal-noise schedule $\{\alpha_t, \sigma_t\}_{t=1,...,T}$, we define a sequence of latent variables $\{\boldsymbol{z}_t\}_{t=0,...,T}$ that satisfies:

$$q(\boldsymbol{z}_t|\boldsymbol{x}) = \mathcal{N}(\boldsymbol{z}_t; \alpha_t \boldsymbol{x}, \sigma_t^2 I), \text{ and } q(\boldsymbol{z}_t|\boldsymbol{z}_s) = \mathcal{N}(\boldsymbol{z}_t; \alpha_{t|s}\boldsymbol{z}_s, \sigma_{t|s}^2 I), \quad (1)$$

where $\boldsymbol{z}_0 = \boldsymbol{x}$, $\alpha_{t|s} = \alpha_t/\alpha_s, \sigma_{t|s}^2 = \sigma_t^2 - \alpha_{t|s}^2\sigma_s^2$, $s < t$. By default, the signal-to-noise ratio (SNR, $\alpha_t^2/\sigma_t^2$) decreases monotonically with t. The model then learns to reverse the process with a backward model $p_\theta(\boldsymbol{z}_{t-1}|\boldsymbol{z}_t)$, which can be re-written as a denoising objective:

$$\mathcal{L}_\theta = \mathbb{E}_{t\sim[1,T],\boldsymbol{z}_t\sim q(\boldsymbol{z}_t|\boldsymbol{x})} \left[\omega_t \cdot \|\boldsymbol{x}_\theta(\boldsymbol{z}_t, t) - \boldsymbol{x}\|_2^2\right],$$

where $\boldsymbol{x}_\theta(\boldsymbol{z}_t, t)$ is a neural network (often a variant of a UNet model (Ronneberger et al., 2015)) that maps a noisy input $\boldsymbol{z}_t$ to its clean version $\boldsymbol{x}$, conditioned on the time step $t$; $\omega_t \in \mathbb{R}^+$ is a loss weighting factor determined by heuristics. In practice, one can reparameterize $\boldsymbol{x}_\theta$ with noise- or v-prediction (Salimans & Ho, 2022) for improved performance. Unlike other generative models like GANs (Goodfellow et al., 2014), diffusion models require repeatedly applying a deep neural network $\boldsymbol{x}_\theta$ in the ambient space as enough computation with global interaction is critical for denoising (Kadkhodaie et al., 2022). This makes it challenging to design efficient diffusion models directly for high-resolution generation, especially for complex tasks like text-to-image synthesis. As common solutions, existing methods have focused on learning hierarchical generation:

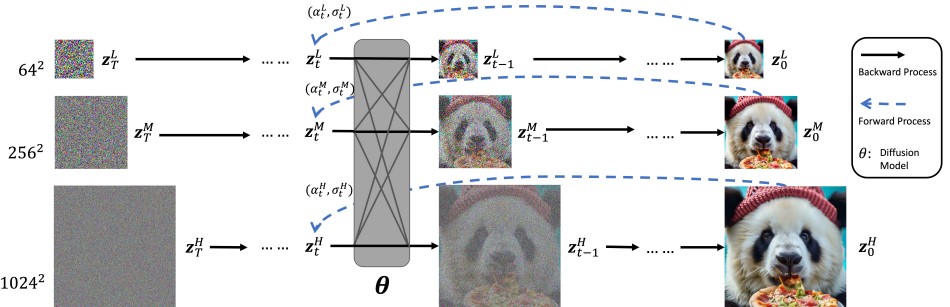

Figure 2: An illustration of Matryoshka Diffusion. $z_t^L$, $z_t^M$ and $z_t^H$ are noisy images at three different resolutions, which are fed into the denoising network together, and predict targets independently.

**Cascaded diffusion (Ho et al., 2022b; Ramesh et al., 2022; Saharia et al., 2022; Ho et al., 2022a; Pernias et al., 2023)** utilize a cascaded approach where a first diffusion model is used to generate data at lower resolution, and then a second diffusion model is used to generate a super-resolution version of the initial generation, taking the first stage generation as conditioning. Cascaded models can be chained multiple times until they reach the final resolution. Ho et al. (2022a); Singer et al. (2022) uses a similar approach for video synthesis as well – models are cascaded from low spatio-temporal resolution to high spatio-temporal resolution. However, since each model is trained separately, the generation quality can be bottlenecked by the exposure bias (Bengio et al., 2015) from imperfect predictions and several models need to be trained corresponding to different resolutions.

**Latent diffusion (LDM, Rombach et al., 2022)** and its follow-ups (Peebles & Xie, 2022; Xue et al., 2023; Podell et al., 2023), on the other hand, handle high-resolution image generation by performing diffusion in the lower resolution latent space of a pre-trained auto-encoder, which is typically trained with adversarial objectives (Esser et al., 2021). This not only increases the complexity of learning, but bounds the generation quality due to the lossy compression process.

**End-to-end models** Recently, several approaches have been proposed (Hoogeboom et al., 2023; Jabri et al., 2022; Chen, 2023) to train end-to-end models directly on high-resolution space. Without relying on separate models, these methods focus on efficient network design as well as shifted noise schedule to adapt high-resolution spaces. Nevertheless, without fully considering the innate structure of hierarchical generation, their results lag behind cascaded and latent models.

## 3 Matryoshka Diffusion Models

In this section, we present Matryoshka Diffusion Models (MDM), a new class of diffusion models that is trained in high-resolution space, while exploiting the hierarchical structure of data formation. MDM first generalizes standard diffusion models in the extended space (§ 3.1), for which specialized nested architectures (§ 3.2) and training procedures (Appendix B) are proposed.

### 3.1 Diffusion Models in Extended Space

Unlike cascaded or latent methods, MDM learns a single diffusion process with hierarchical structure by introducing a multi-resolution diffusion process in an extended space. An illustration is shown in Fig. 2. Given a data point $x \in \mathbb{R}^N$, we define time-dependent latent $z_t = \left[ z_t^1, \ldots, z_t^R \right] \in \mathbb{R}^{N_1 + \ldots N_R}$. Similar to Eq. (1), for each $z_r, r = 1, \ldots, R$:

$$q(z_t^r | x) = \mathcal{N}(z_t^r; \alpha_t^r D^r(x), \sigma_t^{r\,2} I), \tag{2}$$

where $D^r : \mathbb{R}^N \to \mathbb{R}^{N_r}$ is a deterministic "down-sample" operator depending on the data. Here, $D^r(x)$ is a coarse / lossy-compressed version of $x$. For instance, $D^r(.)$ can be `avgpool`$(.)$ for generating low-resolution images.

By default, we assume compression in a progressive manner such that $N_1 < N_2 \ldots < N_R = N$ and $D^R(x) = x$. Also, $\{\alpha_t^r, \sigma_t^r\}$ are the resolution-specific noise schedule. In this paper, we follow Gu et al. (2022) and shift the noise schedule based on the input resolutions. MDM then learns

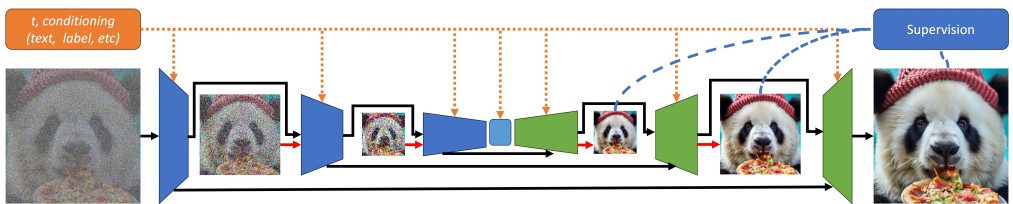

Figure 3: An illustration of the NestedUNet architecture used in Matryoshka Diffusion. We follow the design of Podell et al. (2023) by allocating more computation in the low resolution feature maps (by using more attention layers for example), where in the figure we use the width of a block to denote the parameter counts. Here the black arrows indicate connections inherited from UNet, and red arrows indicate additional connections introduced by Nested UNet.

the backward process $p_\theta(z_{t-1}|z_t)$ with $R$ neural denoisers $x^r_\theta(z_t)$. Each variable $z^r_{t-1}$ depends on all resolutions $\{z^1_t \ldots z^R_t\}$ at time step $t$. During inference, MDM generates all $R$ resolutions in parallel. There is no dependency between $z^r_t$.

Modeling diffusion in the extended space has clear merits: (1) since what we care during inference is the full-resolution output $z^R_t$, all other intermediate resolutions are treated as additional hidden variables $z^r_t$, enriching the complexity of the modeled distribution;(2) the multi-resolution dependency opens up opportunities to share weights and computations across $z^r_t$, enabling us to re-allocate computation in a more efficient manner for both training and inference efficiency.

## 3.2 NESTEDUNET ARCHITECTURE

Similar to typical diffusion models, we implement MDM in the flavor of UNet (Ronneberger et al., 2015; Nichol & Dhariwal, 2021): skip-connections are used in parallel with a computation block to preserve fine-grained input information, where the block consists of multi-level convolution and self-attention layers. In MDM, under the progressive compression assumption, it is natural that the computation for $z^r_t$ is also beneficial for $z^{r+1}_t$. This leads us to propose *NestedUNet*, an architecture that groups the latents of all resolutions $\{z^r_t\}$ in one denoising function as a nested structure, where low resolution latents will be fed progressively along with standard down-sampling. Such multi-scale computation sharing greatly eases the learning for high-resolution generation. A pseudo code for NestedUNet compared with standard UNet is present as follows.

```python
def NestedUNet(z:List[Tensor], h:Tensor=None, o:List[Tensor]=[]):
    # z: list of inputs with increasing resolutions
    # h: output hidden states from previous resolution
    # f_merge, f_skip, f_up, f_down: neural layers
    x = z[-1] if h is None else f_merge(z[-1], h)
    if len(z) > 1:  # move to next resolution
        x = f_skip(x, f_up(NestedUNet(z[:-1], f_down(x), o)))
    else:           # inner UNet at lowest resolution
        x = f_skip(x, f_up(f_mid(f_down(x))))
    o.append(x)     # return results of all resolutions
    return x
```

Aside from the simplicity aspect relative to other hierarchcal approaches, NestedUNet also allows to allocate the computation in the most efficient manner. As shown in Fig. 3, our early exploration found that MDM achieved much better scalibility when allocating most of the parameters & computation in the lowest resolution. Similar findings have also been shown in Hoogeboom et al. (2023).

## 3.3 LEARNING

We train MDM using the normal denoising objective jointly at multiple resolutions, as follows:

$$\mathcal{L}_\theta = \mathbb{E}_{t\sim[1,T]}\mathbb{E}_{z_t\sim q(z_t|x)} \sum_{r=1}^{R} \left[ \omega^r_t \cdot \|x^r_\theta(z_t, t) - D^r(x)\|^2_2 \right], \tag{3}$$

where $\omega_t^r$ is the resolution-specific weighting, and by default we set $\omega_t^r/\omega_t^R = N_R/N_r$.

**Progressive Training** While MDM can be trained end-to-end directly following Eq. (3) which has already shown better convergence than naive baselines, we found a simple progressive training technique, similarly proposed in GAN literature (Karras et al., 2017; Gu et al., 2021), greatly speeds up the training of high-resolution models w.r.t. wall clock time. More precisely, we divide up the training into $R$ phases, where we progressively add higher resolution into the training objective in Eq. (3). This is equivalent to learning a sequence of MDMs on $[z_t^1, \ldots z_t^r]$ until $r$ reaching the final resolution. Thanks to the proposed architecture, we can achieve the above trivially as if progressive growing the networks (Karras et al., 2017). This training scheme avoids the costly high-resolution training from the beginning, and speeds up the overall convergence.

## 4 EXPERIMENTS

MDM is a versatile technique applicableto any problem where input dimensionality can be progressively compressed. We consider two applications beyond class-conditional image generation that demonstrate the effectiveness of our approach – text-to-image and text-to-video generation.

### 4.1 EXPERIMENTAL SETTINGS

**Datasets** In this paper, we only focus on datasets that are publicly available and easily reproducible. For image generation, we performed class-conditioned generation on ImageNet (Deng et al., 2009) at $256 \times 256$, and performed general purpose text-to-image generation using Conceptual 12M (CC12M, Changpinyo et al., 2021) at both $256 \times 256$ and $1024 \times 1024$ resolutions. As additional evidence of generality, we show results on text-to-video generation using WebVid-10M (Bain et al., 2021) at $16 \times 256 \times 256$. We list the dataset and preprocessing details in Appendix F.

The choice of relying extensively on CC12M for text-to-image generative models in the paper is a significant departure from prior works (Saharia et al., 2022; Ramesh et al., 2022) that rely on exceedingly large and sometimes inaccessible datasets, and so we address this choice here. We find that CC12M is sufficient for building high-quality text-to-image models with strong zero-shot capabilities in a relatively short training time (see details in Appendix D.2). This allows for a much more consistent comparison of methods for the community because the dataset is freely available and training time is feasible. We submit here, that CC12M is much more amenable as a common training and evaluation baseline for the community working on this problem.

**Evaluation** In line with prior works, we evaluate our image generation models using Fréchet Inception Distance (FID, Heusel et al., 2017) (ImageNet, CC12M) and CLIP scores (Radford et al., 2021) (CC12M). To examine their zero-shot capabilities, we also report the FID/CLIP scores using COCO (Lin et al., 2014) validation set togenerate images with the CC12M trained models. We also provide additional qualitative samples for image and video synthesis in supplementary materials.

**Implementation details** We implement MDMs based on the proposed NestedUNet architecture, with the innermost UNet resolution set to $64 \times 64$. Similar to Podell et al. (2023), we shift the bulk of self-attention layers to the lower-level ($16 \times 16$) features, resulting in total 450M parameters for the inner UNet. As described in § 3.2, the high-resolution part of the model can be easily attached on top of previous level of the NestedUNet, with a minimal increase in the parameter count. For text-to-image and text-to-video models, we use the frozen FLAN-T5 XL (Chung et al., 2022) as our text encoder due to its moderate size and performance for language encoding. Additionally, we apply two learnable self-attention layers over the text representation to enhance text-image alignment.

For image generation tasks, we experiment with MDMs of $\{64^2, 256^2\}$, $\{64^2, 128^2, 256^2\}$ for $256 \times 256$, and $\{64^2, 256^2, 1024^2\}$, $\{64^2, 128^2, 256^2, 512^2, 1024^2\}$ for $1024 \times 1024$, respectively. For video generation, MDM is nested by the same image $64 \times 64$ UNet with additional attention layers for learning temporal dynamics. The overall resolution is $\{64^2, 16 \times 64^2, 16 \times 256^2\}$. We use bi-linear interpolation for spatial $D^r(.)$, and first-frame indexing for temporal $D^r(.)$. Unless specified, we apply progressive and mixed-resolution training for all MDMs. We use 8 A100 GPUs for ImageNet, and 32 A100 GPUs for CC12M and WebVid-10M, respectively. See Appendices A and B for more implementation hyper-parameters and training details.

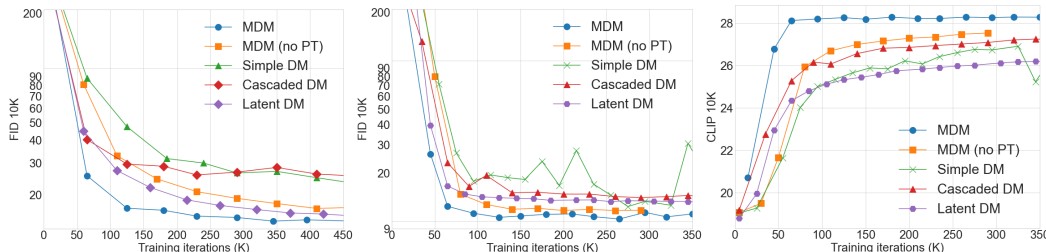

(a) FID (↓) of ImageNet 256 × 256. (b) FID (↓) on CC12M 256 × 256. (c) CLIP (↑) on CC12M 256 × 256.

Figure 4: Comparison against baselines during training. FID (↓) (a, b) and CLIP(↑) (c) scores of samples generated without CFG during training of different class conditional models of ImageNet 256 × 256 (a) and CC12M 256 × 256 (b, c). As can be seen, MDM models that were first trained at lower resolution (200K steps for ImageNet, and 390K for CC12M here) converge much faster.

**Baseline models** Aside from the comparisons with existing state-of-the-art approaches, we also report detailed analysis on MDMs against three baseline models under controlled setup:

1. *Simple DM*: A standard UNet architecture directly applied to high resolution inputs; We also consider the Nested UNet architecture, but ignoring the low resolution losses; Both cases are essentially identical to recent end-to-end diffusion models like Hoogeboom et al. (2023).

2. *Cascaded DM*: we follow the implementation details of Saharia et al. (2022) and train a CDM that is directly comparable with MDM where the upsampler has an identical configuration to our NestedUNet. We also apply noise augmentation to the low resolution conditioning image, and sweep over the optimal noise level during inference.

3. *Latent DM*: we utilize the latent codes derived from the auto-encoders from Rombach et al. (2022),and subsequently train diffusion models that match the dimensions of the MDM UNet.

## 4.2 MAIN RESULTS

**Comparison with baseline approaches** Our comparisons to baselines are shown in Fig. 4. On ImageNet 256 × 256, we select a standard UNet our simple DM baseline. For the Cascaded DM baseline, we pretrain a 64x64 diffusion model for 200K iterations, and apply an upsampler UNet also in the same size. We apply standard noise augmentation and sweep for the optimal noise level during inference time (which we have found to be critical). For LDM experiments, we use pretrained autoencoders from Rombach et al. (2022) which downsamples the input resolution and we use the same architecture for these experiments as our 64x64 low resolution models. For MDM variants, we use a NestedUNet of the same size as the baseline UNet. We experiment with two variants, one trained directly with the multi resolution loss Eq. (3) (denoted as no PT), and another one resuming from the 64x64 diffusion model (ie, progressive training). CC12M 256x256 follows a similar setting, except that we use a single loss NestedUNet as our simple DM architecture. We monitor the FID curve on ImageNet, and the FID and CLIP curves on CC12M.

Table 1: Comparison with literature on ImageNet (FID-50K), and COCO (FID-30K). * indicates samples are generated with CFG. Note existing text-to-image models are mostly trained on much bigger datasets than CC12M.

| Models | FID ↓ |
|---|---|
| **ImageNet** 256 × 256 | |
| ADM (Nichol & Dhariwal, 2021) | 10.94 |
| CDM (Ho et al., 2022b) | 4.88 |
| LDM-4 (Rombach et al., 2022) | 10.56 |
| LDM-4* (Rombach et al., 2022) | 3.60 |
| Ours (cfg=1) | 8.18 |
| Ours (cfg=1.5)* | **3.51** |
| **MS-COCO** 256 × 256 | |
| LDM-8 (Rombach et al., 2022) | 23.31 |
| LDM-8* (Rombach et al., 2022) | 12.63 |
| Dalle-2* (Ramesh et al., 2022) | 10.39 |
| IMAGEN* (Saharia et al., 2021) | 7.27 |
| Ours (cfg=1) | 18.35 |
| Ours (cfg=1.35)* | 13.43 |

Comparing simple DM to MDM, we see that MDM clearly has faster convergence, and reaches better performance in the end. This suggests that the multi resolution diffusion process together with the multi resolution loss effectively improves the models convergence, with negligible added complexities. When following the progressive training schedule, we see that MDM's performance and convergence speed further improves. As a direct comparison, we see that the Cascaded DM baseline significantly underperforms MDM, while both starting from the same 64x64 model. Note

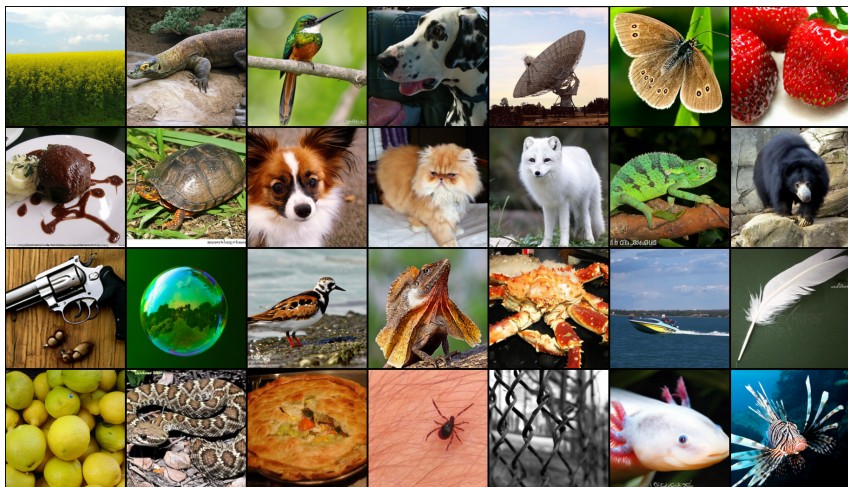

Figure 5: Random samples from our class-conditional MDM trained on ImageNet $256 \times 256$.

that this is remarkable because Cascaded DM has more combined parameters than MDM (because MDM has extensive parameter sharing across resolutions), and uses twice as many inference steps. We hypothesize that the inferior performance of Cascaded DM is largely due to the fact that our 64x64 is not aggressively trained, which causes a large gap between training and inference wrt the conditioning inputs. Lastly, compared to LDM, MDM also shows better performance. Although this is a less direct control as LDM is indeed more efficient due to its small input size, but MDM features a simpler training and inference pipeline.

**Comparison with literature** In Table 1, MDM is compared to existing approaches in literature, where we report FID-50K for ImageNet 256x256 and zero shot FID-30K on MSCOCO. On ImageNet, for which our architecture and hyperparameters are not optimized, MDM is able to achieve competitive FID of 3.51 with CFG. Our FID results comparable to the literature, although MDM is trained on significantly less data than the baselines like Imagen and Dalle-2.

**Qualitative Results** We show random samples from the trained MDMs on for image generation (ImageNet $256 \times 256$, Fig. 5), text-to-image (CC12M, $1024 \times 1024$ Fig. 6) and text-to-video (WebVid-10M, Fig. 7). Despite training on relatively small datasets, MDMs show strong zero-shot capabilities of generating high-resolution images and videos. Note that we use the same training pipelines for all three tasks, indicating its versatile abilities of handling various data types.

### 4.3 ABLATION STUDIES

**Effects of progressive training** We experiment with the progressive training schedule, where we vary the number of iterations that the low-resolution model is trained on before continuing on the target resolution (Fig. 8a). We see that more low resolution training clearly benefits that of the high-resolution FID curves. Note that training on low resolution inputs is much more efficient w.r.t. both memory and time complexity, progressive training provides a straightforward option for finding the best computational trade-offs during training.

**Effects of nested levels** Next, we compare the performance of using different number of nested resolutions with experiments on CC12M. The result is shown in Fig. 8b. We see that increasing from two resolution levels to three consistently improves the model's convergence. It's also worth noting that increasing the number of nesting levels brings only negligible costs.

**CLIP-FID trade-off** Lastly, we show in Fig. 8c the pereto curve of CLIP-FID on the zero-shot evaluation of COCO, achieved by varying the classifier free guidance (CFG) weight. MDM is similarly amendable to CFG as other diffusion model variants. As a comparison, we overlap the same plot reported by Imagen (Figure A.11). We see that Imagen in general demonstrates smaller FID, which we attribute it to higher diversity as a result of training on a large dataset. However, MDM demonstrates strong CLIP score, whereas we have found in practice that such high CLIP scores correlate very well with the visual quality of the generated images.

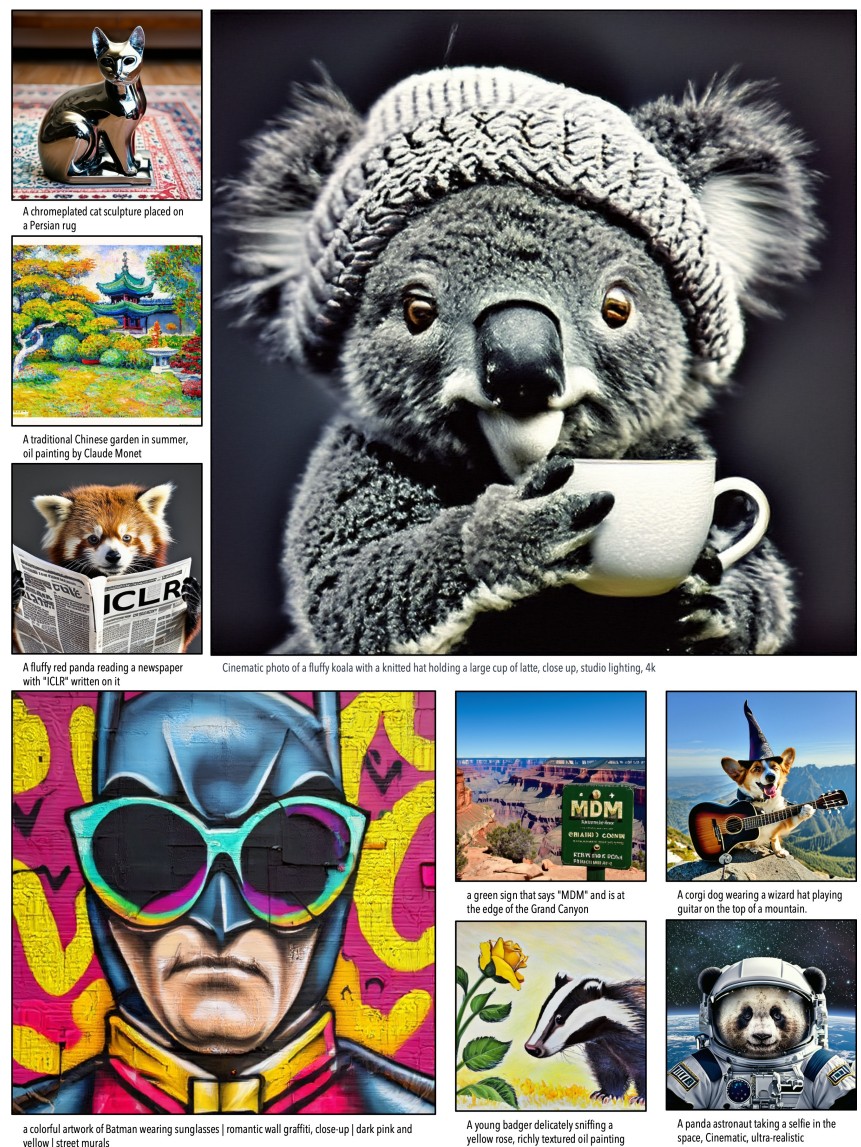

Figure 6: Samples from the model trained on CC12M at $1024^2$ with progressive training.

## 5 RELATED WORK

In addition to diffusion methods covered in § 2, multiscale models have been widely used in image generation and representation learning (Kusupati et al., 2022). A well-known Generative Adversarial Network (GAN) is the LAPGAN model (Denton et al., 2015) which generates lower-resolution images that are subsequently fed into higher-resolution models. Pyramidal Diffusion (Ryu & Ye, 2022), applies a similar strategy with denoising diffusion models. Autoregressive models have also been applied for generation – from early works for images (Van Den Oord et al., 2016; Oord et al., 2016) and videos (Kalchbrenner et al., 2017; Weissenborn et al., 2020), to more recent text-to-image models (Gafni et al., 2022; Yu et al., 2022) and text to video models (Wu et al., 2021; Singer et al., 2022). While earlier works often operate in pixel space, recent works, such as Parti (Yu et al., 2022) and MakeAScene (Gafni et al., 2022) use autoencoders to preprocess images into discrete latent features which can be modeled autoregressively using large sequence-to-sequence models based on transformers. f-DM (Gu et al., 2022) proposed a generalized framework enabling progressive signal transformation across multiple scales, and derived a corresponding de-noising scheduler to transit from multiple resolution stages. This scheduler is employed in our work. Similarly, IHDM (Rissanen et al., 2023) does coarse-to-fine generation implicitly increase the resolution.

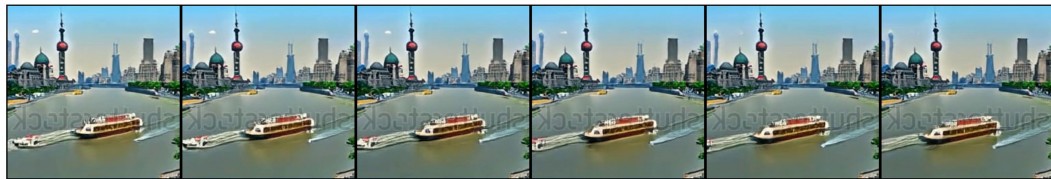

The Bund, Shanghai, with the ship moving on the river, oil painting

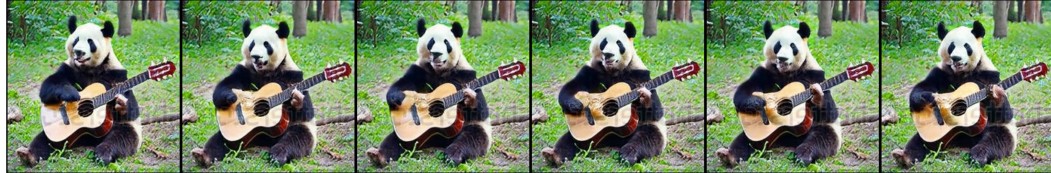

A happy giant panda playing guitar in the forest,day,

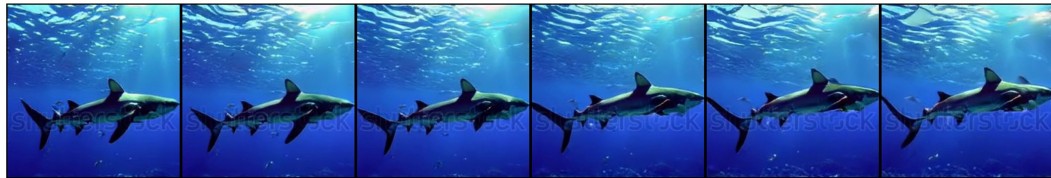

A shark swimming in the ocean

Figure 7: Samples from the model trained on WebVid-10M at $16 \times 256^2$ with progressive training. Videos are subsampled for ease of visualiation.

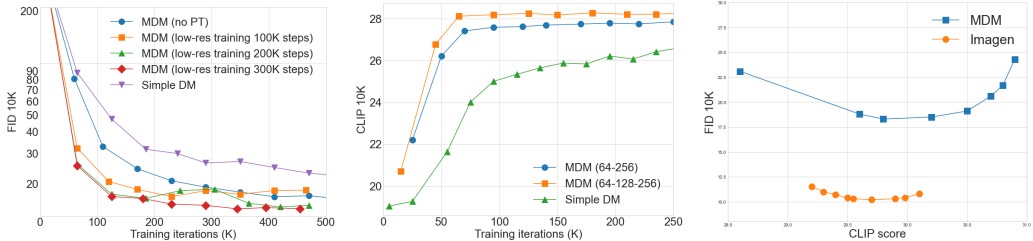

(a) FID ($\downarrow$) on ImageNet $256 \times 256$. (b) CLIP ($\uparrow$) on CC12M $256 \times 256$. (c) Trade-off on COCO $256 \times 256$.

Figure 8: (a) Increasing the number of steps of low resolution training in the progressive training improves results. (b) Larger number of nesting levels on CLIP produces more improvements in speed of convergence and final score (c) FID vs CLIP trade-off seen by varying the weight of CFG (using evaluation on COCO)

## 6  DISCUSSIONS AND FUTURE DIRECTIONS

In this paper we showed that sharing representations across different resolutions can lead to faster training with high quality results, when lower resolutions are trained first. We believe this is because the model is able to exploit the correlations across different resolutions more effectively, both spatially and temporally. While we explored only a small set of architectures here, we expect more improvements can be achieved from a more detailed exploration of weight sharing architectures, and new ways of distributing parameters across different resolutions in the current architecture. Another unique aspect of our work is the use of an augmented space, where denoising is performed over multiple resolutions jointly. In this formulation resolution over time and space are treated in the same way, with the differences in correlation structure in time and space being learned by different parameters of the weight sharing model. A more general way of conceptualizing the joint optimization over multiple resolutions is to decouple the losses at different resolutions, by weighting them differently. It is conceivable that a smooth transition can be achieved from training on lower to higher resolution. We also note that while we have compared our approach to LDM in the paper, these methods are complementary. It is possible to build MDM on top of autoencoder codes. While we are not making the claim that the MDM based models are reaching the SOTA, we leave the evaluation of MDM on large scale dataset and model sizes as future work.

ACKNOWLEDGEMENT

We thank Miguel Angel Bautista, Jason Ramapuram, Alaaeldin El-Nouby, Laurent Dinh, Ruixiang Zhang, Yuyang Wang for their critical suggestions and valuable feedback to this project. We thank Ronan Collobert, David Grangier and Awni Hanun for their invaluable support and contributions to the dataset pipeline.

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

# APPENDIX

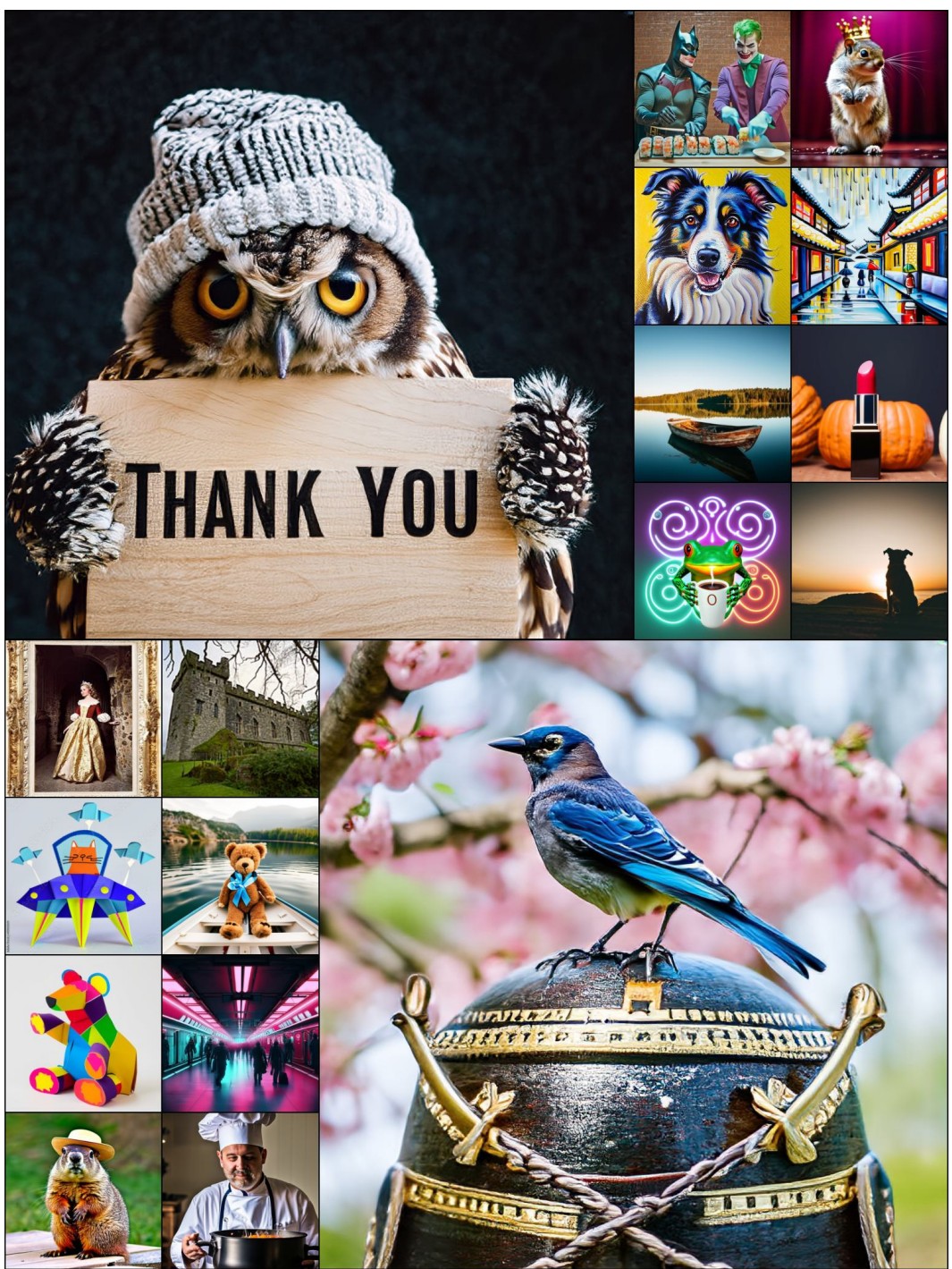

Figure 9: Random samples from MDM trained on CC12M dataset at $256 \times 256$ and $1024 \times 1024$ resolutions. See detailed captions in the Appendix G.

## A ARCHITECTURES

First, we show the following as the core architecture for MDM for the lowest resolution of $64 \times 64$. Following (Podell et al., 2023), we increase the number of self-attention layers for each resnet blocks for $16 \times 16$ computations. To improve the text-image correspondence, we found it useful to apply additional self-attention layers on top of the language model features.

**Base architecture (MDM-S64)**

```
config:
    resolutions=[64,32,16]
    resolution_channels=[256,512,768]
    num_res_blocks=[2,2,2]
    num_attn_layers_per_block=[0,1,5]
    num_heads=8,
    schedule='cosine'
    emb_channels=1024,
    num_lm_attn_layers=2,
    lm_feature_projected_channels=1024
```

Then, we configure the models for $256^2$ and $1024^2$ resolutions in a nested way as follows:

**Nested architecture (MDM-S64S256)**

```
config:
    resolutions=[256,128,64]
    resolution_channels=[64,128,256]
    inner_config:
        resolutions=[64,32,16]
        resolution_channels=[256,512,768]
        num_res_blocks=[2,2,2]
        num_attn_layers_per_block=[0,1,5]
        num_heads=8,
        schedule='cosine'
    num_res_blocks=[2,2,1]
    num_attn_layers_per_block=[0,0,0]
    schedule='cosine-shift4'
    emb_channels=1024,
    num_lm_attn_layers=2,
    lm_feature_projected_channels=1024
```

**Nested architecture (MDM-S64S128S256)**

```
Architecture config (MDM-64,128,256):
    resolutions=[256,128]
    resolution_channels=[64,128]
    inner_config:
        resolutions=[128,64]
        resolution_channels=[128,256]
        inner_config:
            resolutions=[64,32,16]
            resolution_channels=[256,512,768]
            num_res_blocks=[2,2,2]
            num_attn_layers_per_block=[0,1,5]
            num_heads=8,
            schedule='cosine'
        num_res_blocks=[2,1]
        num_attn_layers_per_block=[0,0]
        schedule='cosine-shift2'
    num_res_blocks=[2,1]
    num_attn_layers_per_block=[0,0]
    schedule='cosine-shift4'
    emb_channels=1024,
    num_lm_attn_layers=2,
    lm_feature_projected_channels=1024
```

**Nested architecture (MDM-S64S256S1024)**

```
config:
    resolutions=[1024,512,256]
    resolution_channels=[32,32,64]
    inner_config:
        resolutions=[256,128,64]
        resolution_channels=[64,128,256]
        inner_config:
            resolutions=[64,32,16]
            resolution_channels=[256,512,768]
            num_res_blocks=[2,2,2]
            num_attn_layers_per_block=[0,1,5]
            num_heads=8,
            schedule='cosine'
        num_res_blocks=[2,2,1]
        num_attn_layers_per_block=[0,0,0]
        schedule='cosine-shift4'
    num_res_blocks=[2,2,1]
    num_attn_layers_per_block=[0,0,0]
    schedule='cosine-shift16'
    emb_channels=1024,
    num_lm_attn_layers=2,
    lm_feature_projected_channels=1024
```

**Nested architecture (MDM-S64S128S256S512S1024)**

```
config:
    resolutions=[1024,512]
    resolution_channels=[32,32]
    inner_config:
        resolutions=[512,256]
        resolution_channels=[32,64]
        inner_config:
            resolutions=[256,128]
            resolution_channels=[64,128]
            inner_config:
                resolutions=[128,64]
                resolution_channels=[128,256]
                inner_config:
                    resolutions=[64,32,16]
                    resolution_channels=[256,512,768]
                    num_res_blocks=[2,2,2]
                    num_attn_layers_per_block=[0,1,5]
                    num_heads=8,
                    schedule='cosine'
                num_res_blocks=[2,1]
                num_attn_layers_per_block=[0,0]
                schedule='cosine-shift2'
            num_res_blocks=[2,1]
            num_attn_layers_per_block=[0,0]
            schedule='cosine-shift4'
        num_res_blocks=[2,1]
        num_attn_layers_per_block=[0,0]
        schedule='cosine-shift8'}
    num_res_blocks=[2,1]
    num_attn_layers_per_block=[0,0]
    schedule='cosine-shift16'
    emb_channels=1024,
    num_lm_attn_layers=2,
    lm_feature_projected_channels=1024
```

In addition, we also show the models for video generation experiments, where additional temporal attention layer is performed across the temporal dimension connected with convolution-based re-

sampling. An illustration of the architecture of video modeling is shown in Fig. 10. For ease of visualization, we use 4 frames instead of 16 which was used in our main experiments.

**Nested architecture (MDM-S64T16) for video generation**

```
config:
    temporal_axis=True
    temporal_resolutions=[16,8,4,2,1]
    resolution_channels=[256,256,256,256,256]
    inner_config:
        resolutions=[64,32,16]
        resolution_channels=[256,512,768]
        num_res_blocks=[2,2,2]
        num_attn_layers_per_block=[0,1,5]
        num_heads=8,
        schedule='cosine'
    num_res_blocks=[2,2,2,2,1]
    num_attn_layers_per_block=[0,0,0,0,0]
    num_temporal_attn_layers_per_block=[1,1,1,1,0]
    schedule='cosine-shift4'
    emb_channels=1024,
    num_lm_attn_layers=2,
    lm_feature_projected_channels=1024
```

**Nested architecture (MDM-S64T16S256) for video generation**

```
config:
    resolutions=[256,128,64]
    resolution_channels=[64,128,256]
    inner_config:
        temporal_axis=True
        temporal_resolutions=[16,8,4,2,1]
        resolution_channels=[256,256,256,256,256]
        inner_config:
            resolutions=[64,32,16]
            resolution_channels=[256,512,768]
            num_res_blocks=[2,2,2]
            num_attn_layers_per_block=[0,1,5]
            num_heads=8,
            schedule='cosine'
        num_res_blocks=[2,2,2,2,1]
        num_attn_layers_per_block=[0,0,0,0,0]
        num_temporal_attn_layers_per_block=[1,1,1,1,0]
        schedule='cosine-shift4'
    num_res_blocks=[2,2,1]
    num_attn_layers_per_block=[0,0,0]
    schedule='cosine-shift16'
    emb_channels=1024,
    num_lm_attn_layers=2,
    lm_feature_projected_channels=1024
```

## B  TRAINING DETAILS

For all experiments, we share all the following training parameters except the `batch_size` and `training_steps` differ across different experiments.

```
default training config:
    optimizer='adam'
    adam_beta1=0.9
    adam_beta2=0.99
    adam_eps=1.e-8
    learning_rate=1e-4
    learning_rate_warmup_steps=30_000
    weight_decay=0.0
```

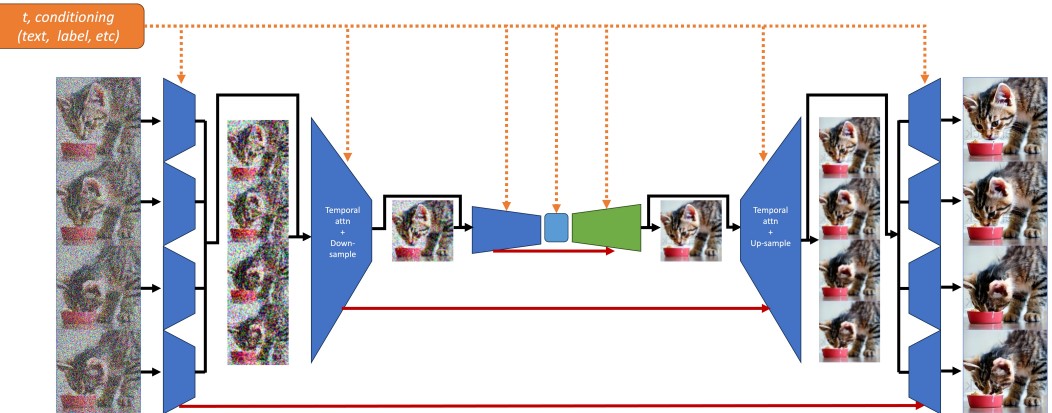

Figure 10: An illustration of the NestedUNet architecture used in Matryoshka Diffusion for video generation. We allocate more computation in the low resolution feature maps, and use additional temporal attention layers to aggregate information across frames.

```
gradient_clip_norm=2.0
ema_decay=0.9999
mixed_precision_training=bp16
```

For ImageNet experiments, the progressive training setting is set default without specifying:

```
progressive training config:
    target_resolutions=[64,256]
    batch_size=[512,256]
    training_steps=[300K,500K]
```

For text-to-image generation on CC12M, we test on both $256 \times 256$ and $1024 \times 1024$ resolutions, while each resolution two types of models with various nesting levels are tested. Note that, the number of progressive training stages is not necessarily the same the actual nested resolutions in the model. For convenience, we always directly initialize the training of $1024 \times 1024$ training with the trained model for $256 \times 256$. Therefore, we can summarize all experiments into one config:

```
progressive training config:
    target_resolutions=[64,256,1024(optional)]
    batch_size=[2048,1024,768]
    training_steps=[500K,500K,100K]
```

Similarly, we list the training config for the video generation experiments as follows.

```
progressive training config:
    target_resolutions=[64,16x64,16x256]
    batch_size=[2048,512,128]
    training_steps=[500K,500K,300K]
```

## C    INFERENCE DETAILS

In Fig. 11, we demonstrate the typical sampling process of a trained MDM. We start by sampling independent Gaussian noises for each resolution which gives us $\{z_T^r\}_{r=1,...,R}$. We then pass all the noisy images to the Nested UNet in parallel, which yields the denoised outputs $\{f(z)_T^r\}_{r=1,...,R}$. We then perform one step of denoising for each $\{z_T^r\}$ with $\{f(z)_T^r\}$, following the procedure of a standard diffusion model, and this gives us $\{z_{T-1}^r\}_{r=1,...,R}$. In practice, we set the number of inference steps as 250 and use v-prediction (Salimans & Ho, 2022) as our model parameterization. Similar to Saharia et al. (2022), we apply "dynamic thresholding" to avoid over-satruation problem in the pixel predictions.

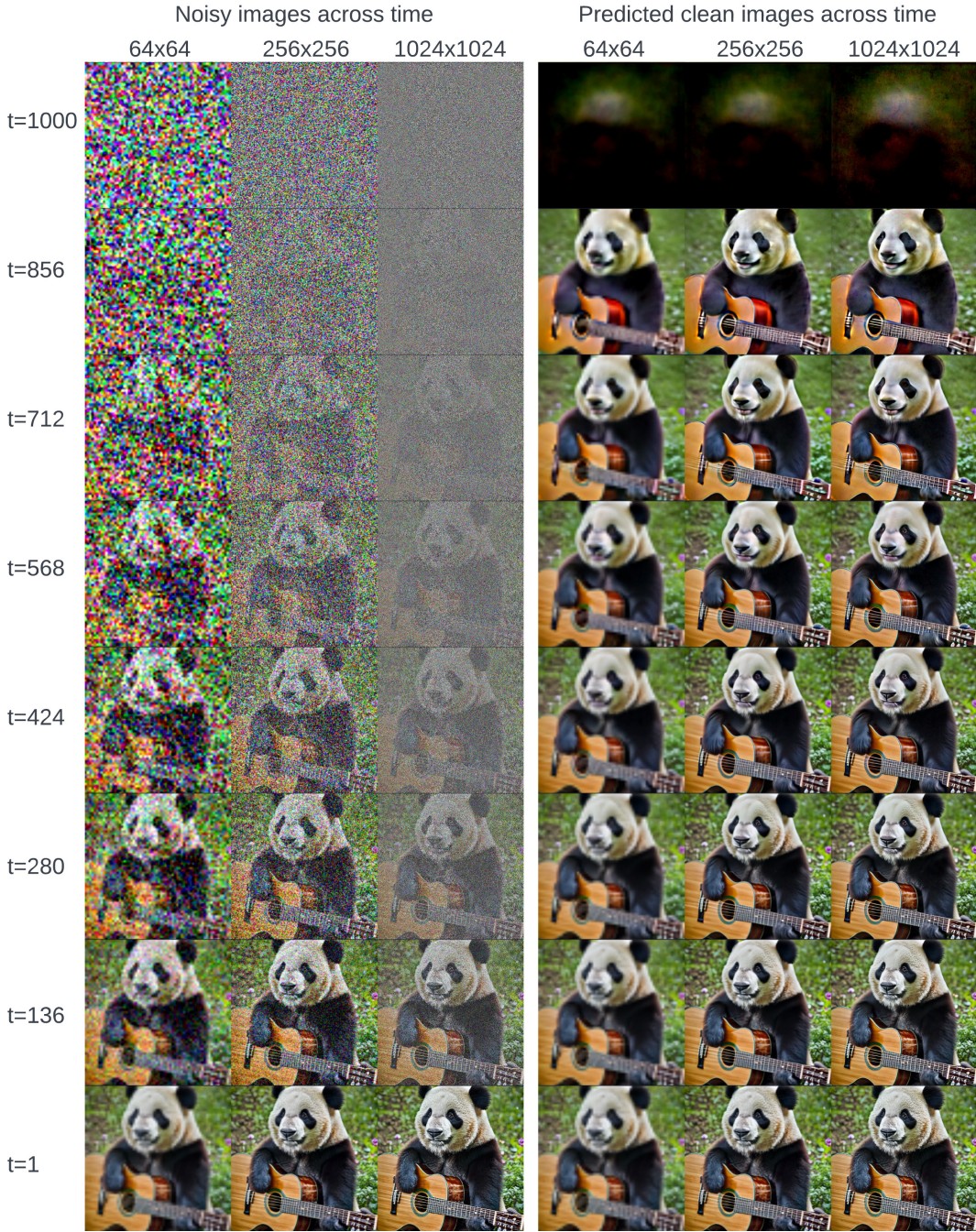

Figure 11: An example of the inference process of MDM for text-to-image generation at $1024 \times 1024$ with three levels. The text caption is "a panda playing guitar in a garden."

## D  ADDITIONAL ABLATIONS

### D.1  UNET VS NESTED UNET

As mentioned in Sec. 4, UNet and Nested UNet are very similar in capacity and they yield near identical performances when treated as standalone architectures. We verify this in Figure 12, where we train a standard diffusion with a UNet and Nested UNet on ImageNet 256x256. We see that they indeed yield tightly coupled FID curves. In addition, we also measure their efficiency during both

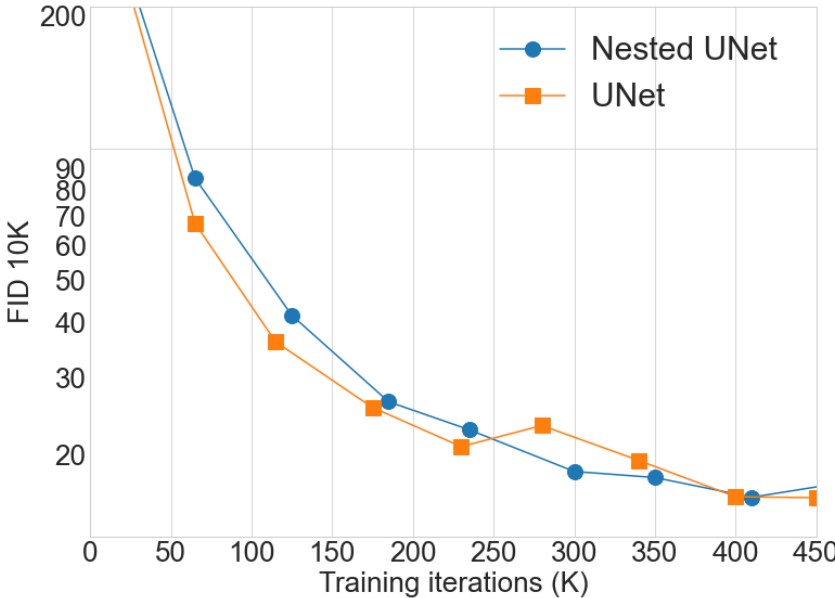

Figure 12: Comparison between standard UNet and Nested UNet on ImageNet 256x256. As expected, they share similar performance, which confirms that Nested UNet as a standalone architecture (with the multi resolution loss and progressive) does not offer immediate advantage over UNet.

Table 2: Speed comparison of UNet and Nested UNet on ImageNet 256x256. Speed is measured on 8xA100 GPUs with a batch size of 128 for training and 256 for inference.

|  | Training (sec/iter) | Inference (sec/step) |
|---|---|---|
| UNet | 0.50 | 0.40 |
| Neseted UNet | 0.52 | 0.41 |

training and inference, as shown in Table 2. It's evident that they have near identical computation costs.

### D.2 TRAINING COSTS

For all three datasets/tasks, namely ImageNet, CC12M and WebVid-10M, our best results are obtained on 4x8 A100 GPUs, for which we here report the total training costs. The results can be seen in Table 3.

### D.3 WALL CLOCK TIME

To complete the comparison in Figure 4 (a), we also provide the time efficiency measurements for each of the baseline runs (only ImageNet experiments are shown, and the comparison is similar on CC12M). This is shown in Table 4.

Table 3: Training cost for our main experiments. INet: ImageNet; CC: CC12M; Vid: WebVid-10M. $\rightarrow$ denotes a progressive training stage, eg, CC 64 $\rightarrow$ 256 refers to training a 256x256 model from a 64x64 model on CC12M. All experiments use 4x8 A100 GPUs.

|  | INet 64 | INet 64 →256 | CC 64 | CC 64 → 256 | CC 256→ 1024 | CC 64 → Vid 16x64 | Vid 16x64 → 16x256 |
|---|---|---|---|---|---|---|---|
| Sample/GPU | 64 | 32 | 64 | 32 | 12 | 8 | 4 |
| Iter(K) | 100 | 375 | 500 | 500 | 100 | 500 | 300 |
| Sec/Iter | 1.01 | 1.21 | 1.14 | 1.38 | 1.88 | 1.17 | 2.34 |

Table 4: Training speed for MDM and baseline models in Figure 4 (a). MDM, simple DM and Cascaded DM have practically the same training speed (and the difference is mostly due to variances in the hardware), while LDM is considerably more efficient.

|  | MDM | MDM (no PT) | Simple DM | Cascaded DM | Latent DM |
|---|---|---|---|---|---|
| Sec/Iter | 1.00 | 1.01 | 1.03 | 0.96 | 0.74 |

# E  BASELINE DETAILS

## E.1  CASCADED DIFFUSION MODEL

For our cascaded diffusion baseline models, we closely follow the guidelines from (Ho et al., 2022b) while making it directly comparable to our models. In particular, our cascaded diffusion models consist of two resolutions, 64x64 and 256x246. Here the 64x64 resolution models share the same architecture and training hyper parameters as MDM. For the upsampler network from 64x64 to 256x256, we upsample the 64x64 conditioning image to 256x256 and concatenate it with the 256x256 noisy inputs. Noise augmentation is applied by applying the same noise scheduels on the upsampled conditioning images, as suggested in (Saharia et al., 2022). All the cascaded diffusion models are trained with 1000 diffusion steps, same as the MDM models.

During inference, we sweep over the noise level used for the conditioning low resolution inputs in the range of $\{1, 100, 500, 700, 1000\}$, similar to Saharia et al. (2022). We found that a relatively high conditioning noise level (500, or 700) is needed for our cascaded models to perform well.

## E.2  LATENT DIFFUSION MODEL

For the LDM experiments we used pretrained encoders from `https://github.com/CompVis/latent-diffusion` (Rombach et al., 2022). The datasets were preprocessed using the autoencoders, and the codes from the autoencoders were modeled by our baseline U-Net diffusion models. For generation, the codes were first generated from the diffusion models, and the decoder of the autoencoders were then used to convert the codes into the images, at the end of diffusion.

In order to follow a similar spatial reduction to our MDM-S64S256 model we reduced the 256x256 images to codes at 64x64 resolution for the Imagenet experiments, using the KL-F4 model from `https://ommer-lab.com/files/latent-diffusion/kl-f4.zip` and we then trained our MDM-S64 baseline model on these spatial codes. However, for the text-to-image diffusion experiments on CC12M we found that the model performed better if we used the 8x downsampling model (KL-F8) – from `https://ommer-lab.com/files/latent-diffusion/kl-f8.zip`. However, since this reduced the resolution of the input to our UNet model, we modified the MDM-S64 model to not perform downsampling after the first ResNet block to preserve a similar computational footprint (and this modification also performed better). The training of the models was performed using the same set of hyperparameters as our baseline models.

# F  DATASETS

**ImageNet (Deng et al., 2009, `https://image-net.org/download.php`)**  contains 1.28M images across 1000 classes. We directly merge all the training images with class-labels. All images are resized to $256^2$ with center-crop. For all ImageNet experiments, we did not perform cross-attention, and fuse the label information together with the time embedding. We did not drop the labels for training both MDM and our baseline models. FID is computed on 50K sampled images against the entire training set images with randomly sampled class labels.

**CC12M (Changpinyo et al., 2021, `https://github.com/google-research-datasets/conceptual-12m`)**  is a dataset with about 12 million image-text pairs meant to be used for vision-and-language pre-training. As mentioned earlier, we choose CC12M as our main training set considering its moderate size for building high-quality text-to-image models with good zero-shot capabilities, and the whole dataset is freely available with less concerning issues like privacy. In this paper, we take all text-image pairs as our dataset set for text-to-image generation. More specifically,

we randomly sample $1/1000$ of pairs as the validation set where we monitor the CLIP and FID scores during training, and use the remaining data for training. Each image by default is center-cropped and resized to desired resolutions depending on the tasks. No additional filtering or cleaning is applied.

**WebVid-10M (Bain et al., 2021, `https://maxbain.com/webvid-dataset`)** is a large-scale dataset of short videos with textual descriptions sourced from stock footage sites. The videos are diverse and rich in their content. Following the preprocessing steps of Guo et al. (2023)[1], we extract each file into a sequence of frames, and randomly sample images every 4 frames to create a 16 frame long clip from the original video. Horizontal flip is applied as additional data augmentation. As the initial exploration of applying MDM on videos, we only sample one clip for each video, and training MDM on the extracted video clips.

## G  ADDITIONAL EXAMPLES

We provide additional qualitative samples from the trained MDMs for ImageNet $256 \times 256$ (Figs. 13 to 15), text-to-image $256 \times 256$ and $1024 \times 1024$ (Figs. 9 and 16 to 18), and text-to-video $16 \times 256 \times 256$ (Fig. 19) tasks.

In particular, the prompts for Fig. 9 are given as follows:

*a fluffy owl with a knitted hat holding a wooden board with "Thank You" written on it* ($1024 \times 1024$),

*batman and Joker making sushi together*,

*a squirrel wearing a crown on stage*,

*an oil painting of Border Collie*,

*an oil painting of rain at a traditional Chinese town*,

*a broken boat in a peacel lake*, *a lipstick put in front of pumpkins*,

*a frog drinking coffee , fancy digital Art*,

*a lonely dog watching sunset*,

*a painting of a royal girl in a classic castle*,

*a realistic photo of a castle*,

*origami style, paper art, a fat cat drives UFO*,

*a teddy bear wearing blue ribbon taking selfie in a small boat in the center of a lake*,

*paper art, paper cut style, cute bear*,

*crowded subway, neon ambiance, abstract black oil, gear mecha, detailed acrylic, photorealistic*,

*a groundhog wearing a straw hat stands on top of the table*,

*an experienced chief making Frech soup in the style of golden light*,

*a blue jay stops on the top of a helmet of Japanese samurai, background with sakura tree* ($1024 \times 1024$).

---

[1]`https://github.com/guoyww/AnimateDiff/blob/main/animatediff/data/dataset.py`

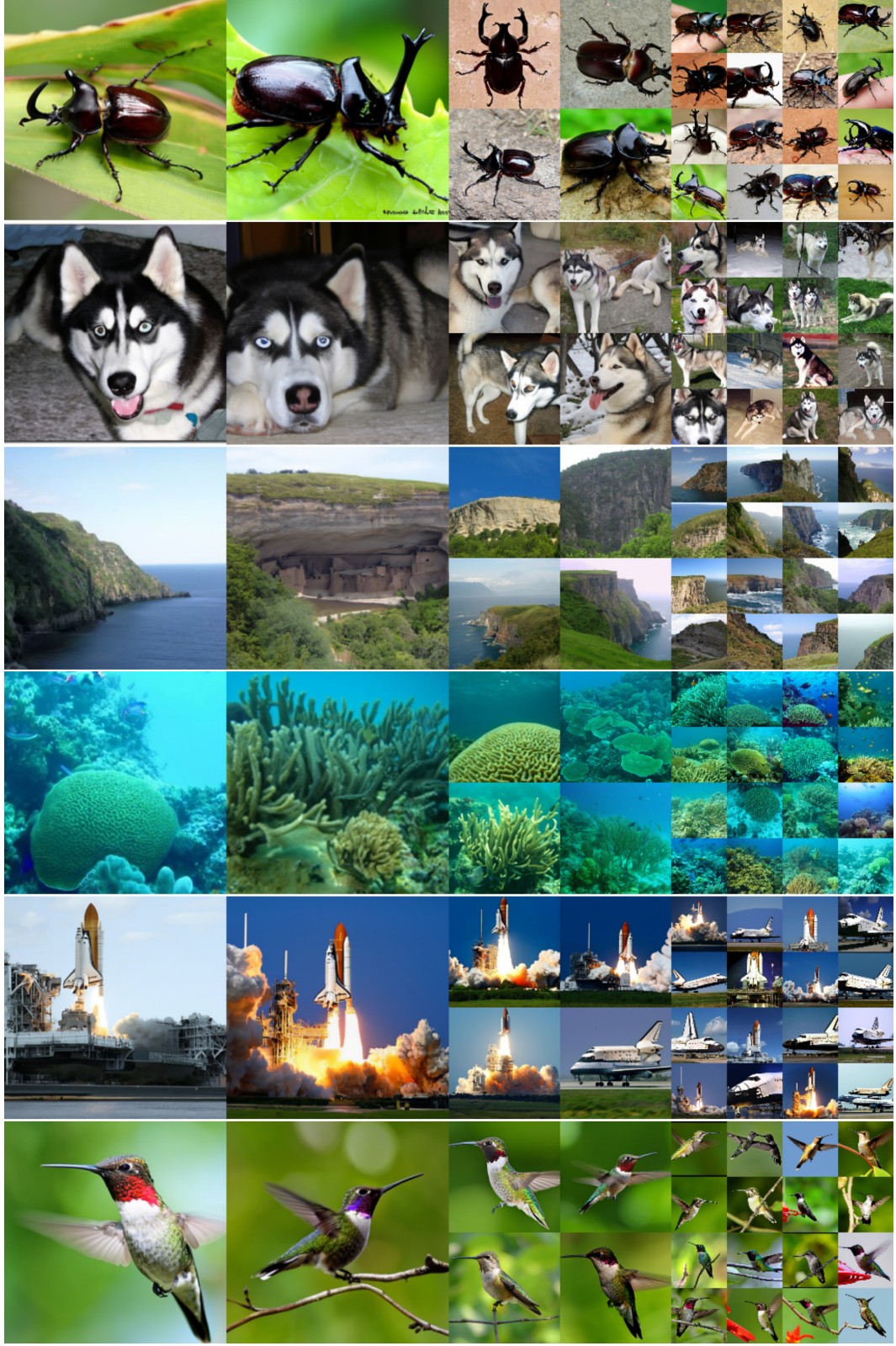

Figure 13: Uncurated samples from MDM trained on ImageNet $256 \times 256$ with the guidance weight 2.5 for labels of *"srhinoceros beetle"*, *"Siberian husky"*, *"cliff, drop, drop-off"*, *"coral reef"*, *"space shuttle"*, *"hummingbird"*.

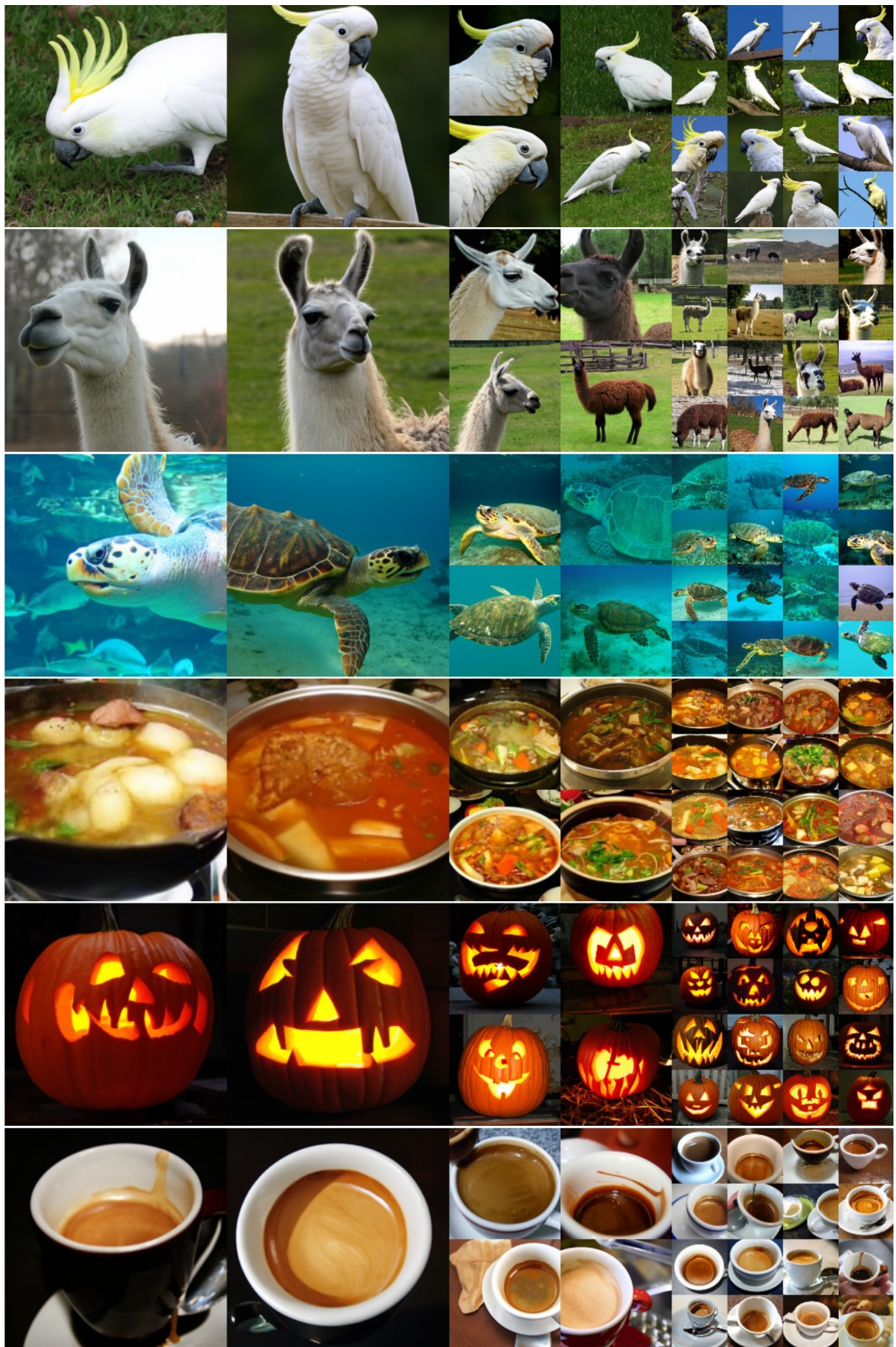

Figure 14: Uncurated samples from MDM trained on ImageNet $256 \times 256$ with the guidance weight 2.5 for labels of *"sulphur-crested cockatoo, Kakatoe galerita, Cacatua galerita"*, *"llama"*, *"loggerhead, loggerhead turtle, Caretta caretta"*, *"hot pot, hotpot"*, *"jack-o'-lantern"*, *"espresso"*.

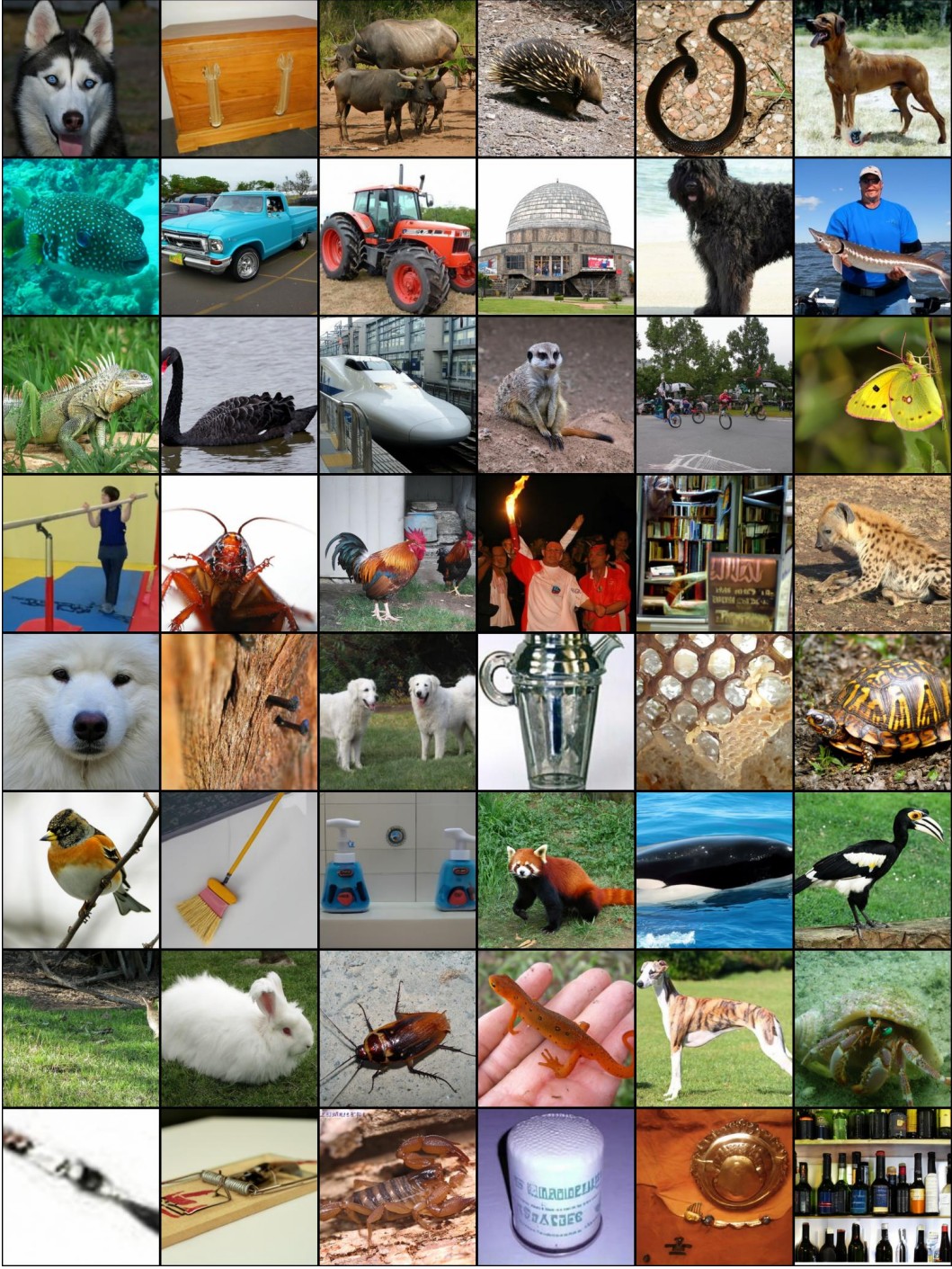

Figure 15: Random samples from the trained MDM on ImageNet $256 \times 256$ given random labels. The guidance weight is set 2.0.

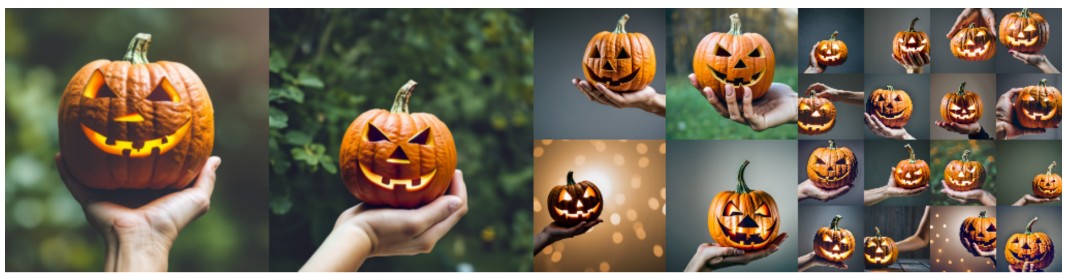

cinematic photo of a hand holding a jack-o'-lantern

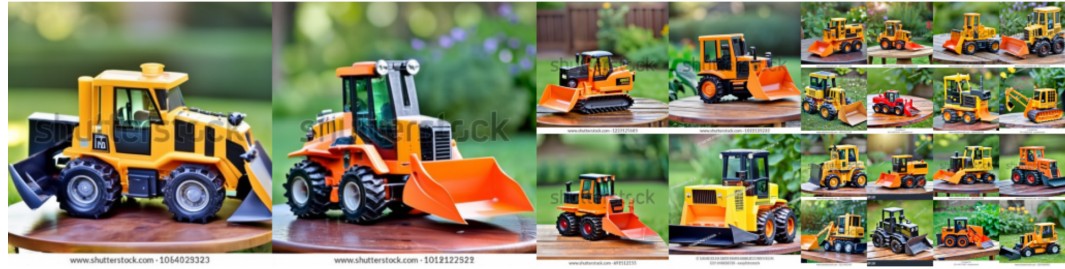

a toy bulldozer on a small table in a garden

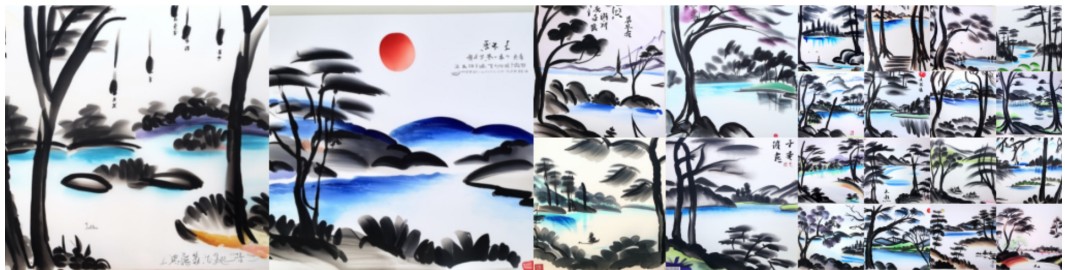

a Chinese ink painting of trees near a peaceful lake

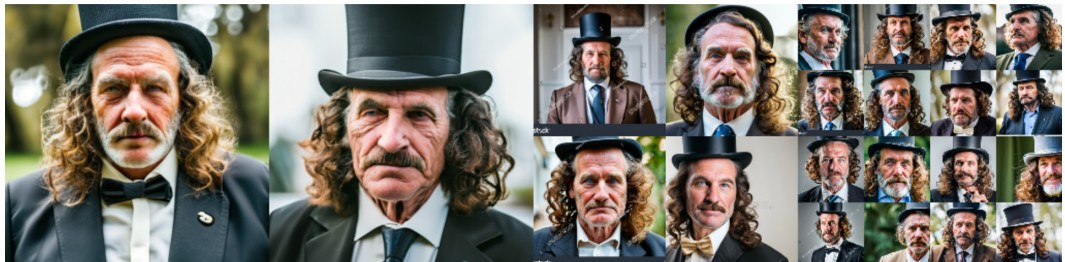

cinematic photo of an old man with long curly hair, blue eyes and wearing a tophat

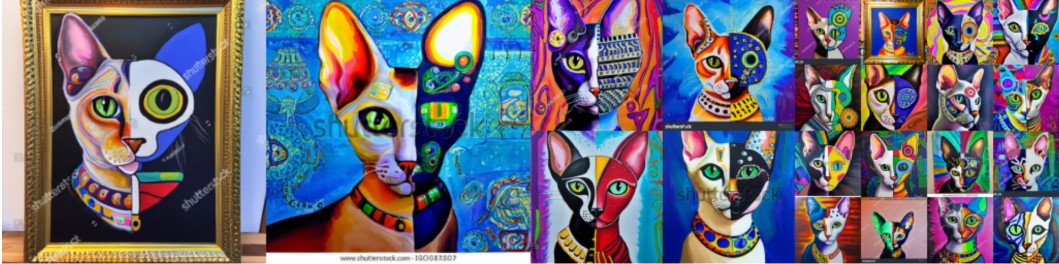

Vbrant portrait painting of an Egyptian cat with a robotic half face

Figure 16: Uncurated samples from the trained MDM on CC12M $256 \times 256$ given various prompts. The guidance weight is set 7.0.

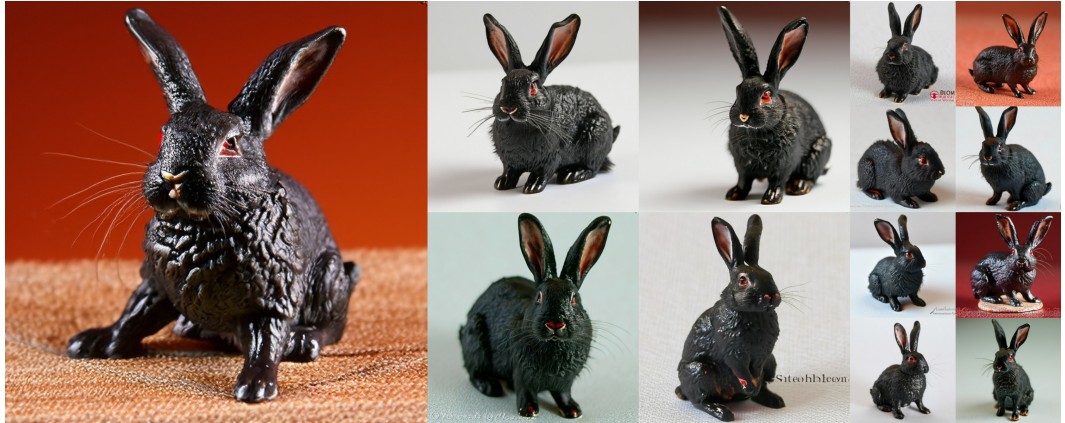

80mm resin detailed miniature of fluffy black devil rabbit, hell, red eyes, Product Introduction Photos

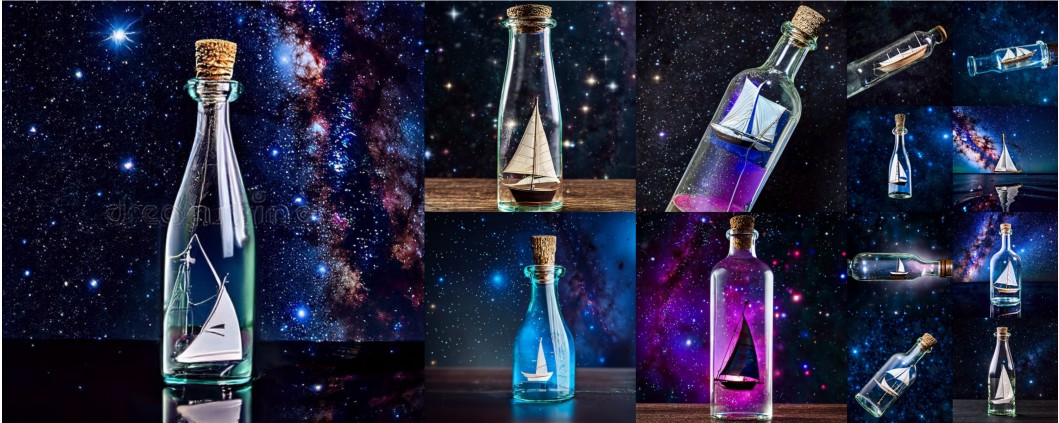

A glass bottle containing a sailboat floats through the galaxy

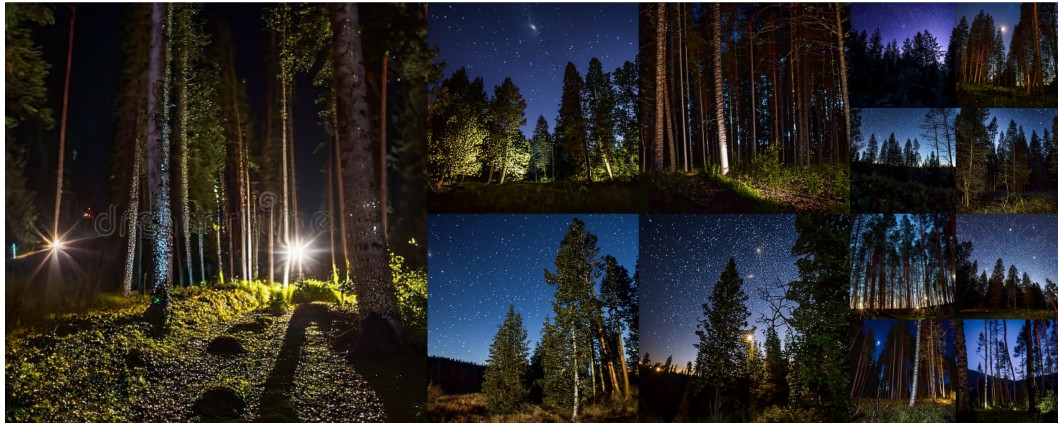

The edge of the Russian forest in summer at night

Figure 17: Random samples from the trained MDM on CC12M $1024 \times 1024$ given various prompts. The guidance weight is set 7.0.

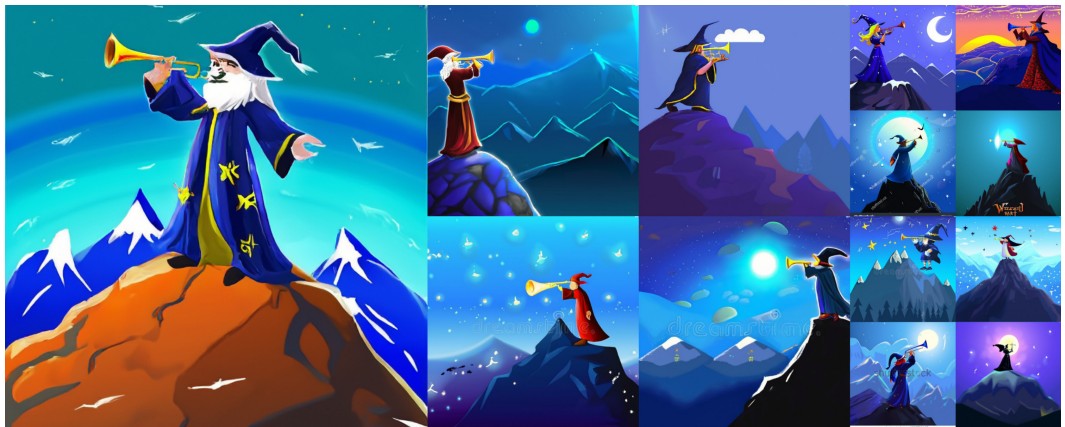

Concept art of a wizard blowing the trumpet on the top of a mountain

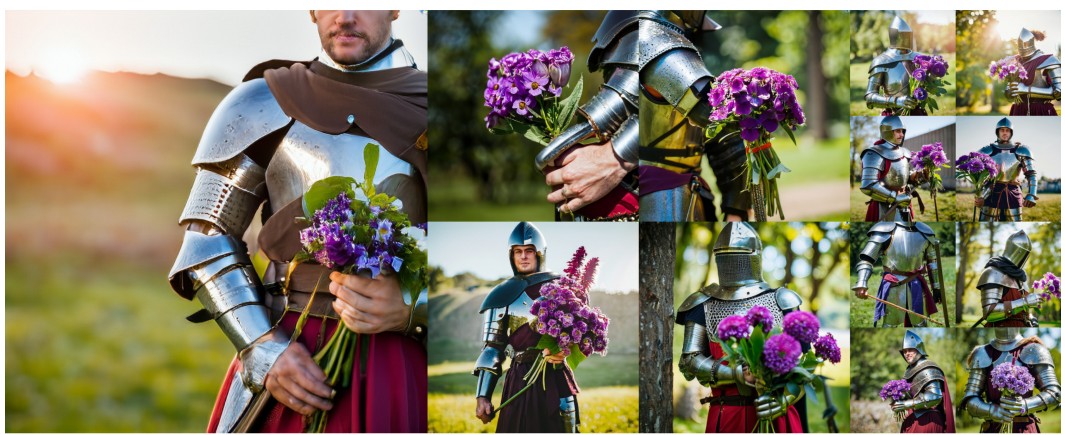

a medieval knight in the daylight with intricate armor details, holding a blooming violet bouquet in his hand. sunshine casts shadows

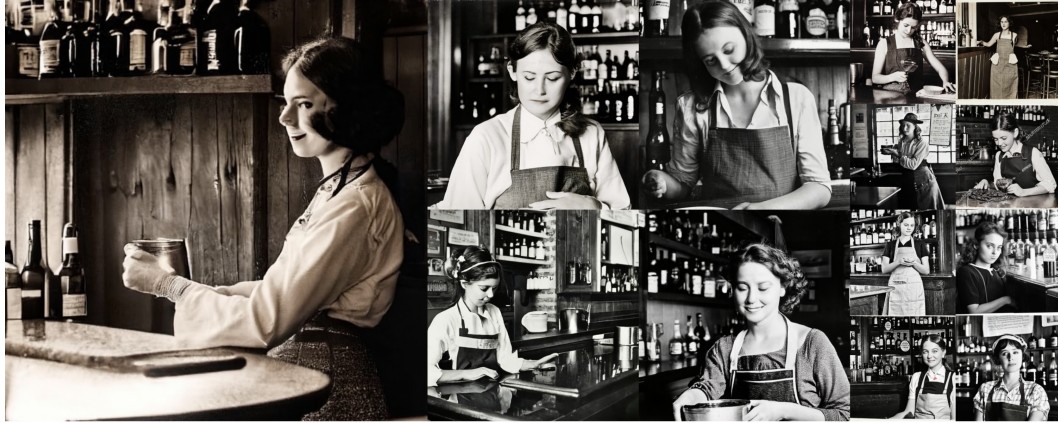

Old photo of a young girl in 1890s in an apron works at a local bar

Figure 18: Random samples from the trained MDM on CC12M $1024 \times 1024$ given various prompts. The guidance weight is set 7.0.

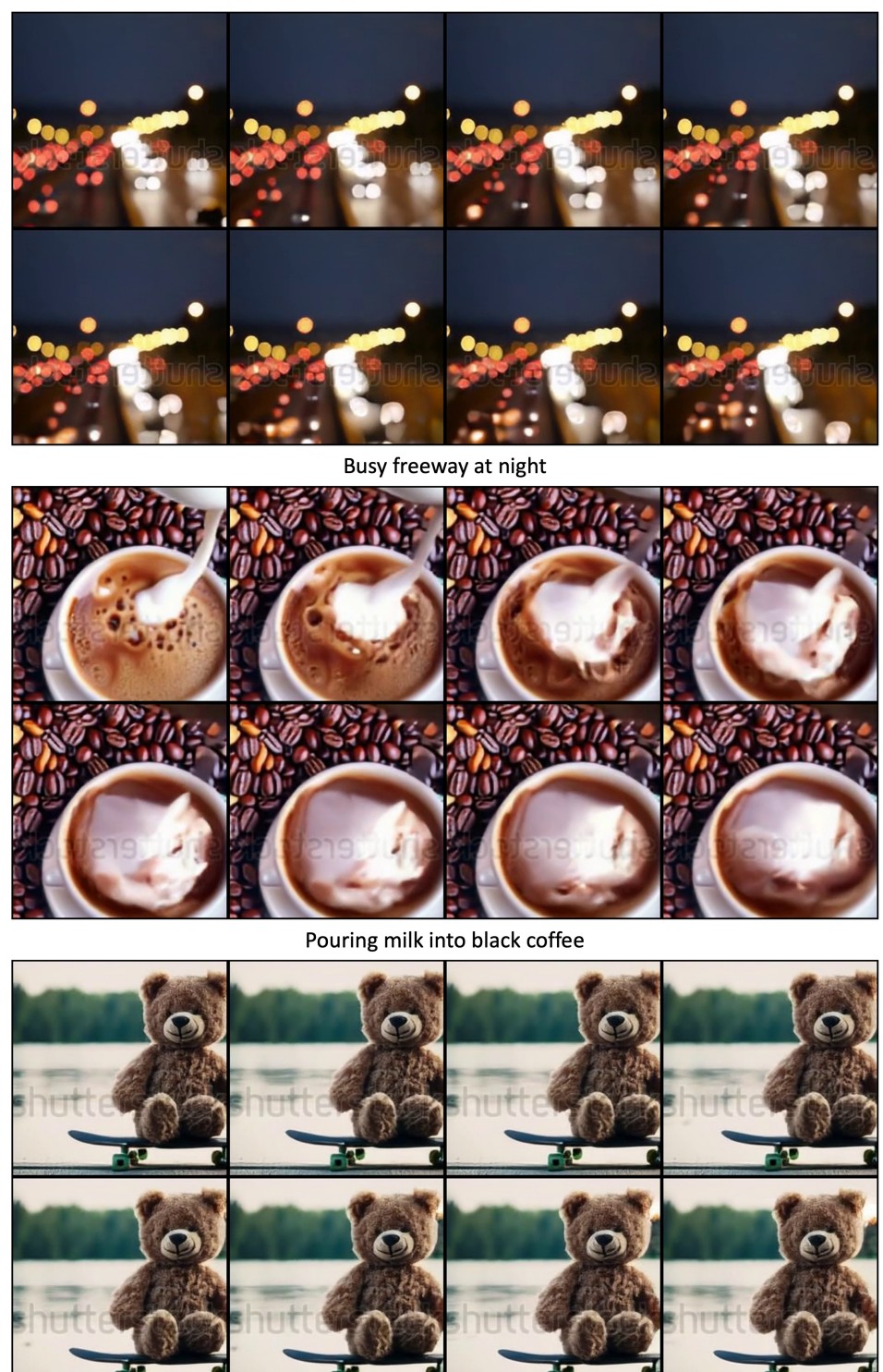

Busy freeway at night

Pouring milk into black coffee

A teddy bear riding a skateboard by the lake

Figure 19: Random samples from the trained MDM on WebVid $16 \times 256 \times 256$ given various prompts. The guidance weight is set 7.0.

