# OpenReview forum: "Matryoshka Diffusion Models"
_ICLR.cc/2024/Conference — ICLR 2024 poster_

### Official Review · Reviewer_cN8u · 2023-10-30

**Soundness:** 2 fair
**Presentation:** 3 good
**Contribution:** 2 fair
**Rating:** 5
**Confidence:** 4

**Summary:**

This paper proposes Matryoshka Diffusion Models for high-resolution image and video synthesis. Unlike conventional approaches that use either cascade models or latent diffusion models with an additional autoencoder, Matryoshka Diffusion Models uses a diffusion process that denoises the multi-resolution input jointly, where such a process can be trained progressively and improves the optimization efficiency significantly. The paper shows the effectiveness of the method on popular image generation and video generation tasks and verifies the training efficiency of the method compared with existing diffusion model variants.

**Strengths:**

- The paper is generally well-written and easy to follow.
- The paper is well-motivated.
- The paper conducts experiments with various datasets, including ImageNet, MSCOCO, and WebVid-10M.
- The high-resolution image generation results are quite impressive.

**Weaknesses:**

- The paper lacks an analysis on "comparison with literature". The paper simply states the result is comparable to other baselines, but the results show a clear gap (e.g., FID 3.60 (LDM) while 6.62 (MDM) on ImageNet 256x256). In this respect, the authors should provide an extensive analysis and reasons why the performance is worse than the baselines, not just saying the proposed method shows comparable performance.
- To verify the "faster convergence", I think x-axis in Figure 4 should be wall-clock time rather than training iterations. Otherwise, I think the authors should provide time/iteration for each baseline used for the evaluation.
- Some important implementation details are missing: learning rate, batch size, model configurations, etc.
- Missing quantitative evaluation on text-to-video generation compared with existing baselines.
- No video files included for illustrating text-to-video generation results.

**Questions:**

- Why some points in Figure 4 (e.g., after 200K of Latent DM in Figure 4(a)) are missing?

---

> ### Author Response · Authors · 2023-11-21
>
> We thank the reviewer for acknowledging the motivation and the results of our paper. We also appreciate your constructive feedbacks. We answer your questions below, and we will provide more details with the updated draft.
>
>
> > The paper lacks an analysis on "comparison with literature"
>
> Our design for the ImageNet experiments are mostly focused on the control experiments, and we have directly borrowed the same architecture and hyper parameters from our text2image experiments. Also, due to computational demands, we did not train our models with a lot of iterations. This means that we have not optimized our FID results on ImageNet, and Table 1 is a rather a pessimistic estimate of our model’s performance. We are working on improving the results, and will update Table 1.
>
>
> > To verify the "faster convergence", I think x-axis in Figure 4 should be wall-clock time rather than training iterations. Otherwise, I think the authors should provide time/iteration for each baseline used for the evaluation.
>
>
> Thanks for the suggestion, we will provide the training cost details in the updated draft. Roughly speaking, the cost per iteration is near identical between single diffusion, MDM and CDM for the high resolution training, all are more costly that LDM.
>
>
> > Some important implementation details are missing: learning rate, batch size, model configurations, etc.
>
>
> We will update the draft for the implementation details. Our design for all the experiments have largely been using the same set of hyper parameters regarding lr, architecture etc, and we only tune the batch size to fit the GPU memory.
>
>
> > Missing quantitative evaluation on text-to-video generation compared with existing baselines.
>
> Due to resource constraints, the experimental results on the text-to-video tasks are preliminary, and our goal is to demonstrate qualitatively that the same idea behind MDM also applies to video generation. There is great interest to us to further improve the video generation’s quality and compare it with state of the art methods, which we leave as future work.
>
>
> > No video files included for illustrating text-to-video generation results.
>
> We will include video results in the supplementary material.
>
>
> > Why some points in Figure 4 (e.g., after 200K of Latent DM in Figure 4(a)) are missing?
>
>
> The LDM experiments were not finished at the point of submission, but we will upload a complete comparison for LDM in Figure 4.

---

> > ### Comment · Reviewer_cN8u · 2023-11-22
> > **Response**
> >
> > Thanks for the detailed response. Since the draft is not updated until now and many evaluations had not been completed in the submission, it is difficult for me to raise the score. I will retain my score.

---

> > > ### Author Response · Authors · 2023-11-22
> > > **The draft will be updated soon as we retrained new models to resolve your concerns**
> > >
> > > Thanks for your comments! We are currently waiting for the last number and will update the draft before the rebuttal deadline as soon as possible. We will provide more detailed responses if you have additional questions and concerns on the evaluations. Please check our draft again if there is a chance. Thanks for your patience!

---

> > > > ### Author Response · Authors · 2023-11-23
> > > > **Draft updated**
> > > >
> > > > Thank you again for your prompt responses. We have updated the draft, with the following changes relevant to your concerns.
> > > >
> > > > 1. **ImageNet Results and comparison to literature**. We have updated Table 1 with much improved FID results on ImageNet. As stated before, we did not optimize our hyper parameters specifically for ImageNet, and also was not training the models long enough. During the rebuttal period, we have tried to increase the training batch size, and also fixed our previous way of performing CFG (we were not properly dropping labels during training in the previous version). After these simple changes, we see drastic improves in the FID numbers which now match the baseline results in Table 1. We are optimistic that there is yet more room to improve with some more hyper parameter tuning. We have also added comparison to the Imagen paper's FID-CLIP plot in Figure 8 (c). More discussions are also included in the paper wrt these updates, see Sec 4.2, 4.3.
> > > >
> > > > 2. **Wallclock time for Figure 4**. We have included the time measurement for models in Figure 4, see Table 4. Essentially, MDM, simple diffusion and Cascaded diffusion have near identical training costs due to the similarity of Nested UNet, UNet, and the upsampling UNet. LDM is considerably faster than the other three due to its lower input dimensionality. Combining Figure 4 and Table 4, we can see that MDM provides a better computational efficiency than simple diffusion and Cascaded diffusion in our controlled setting.
> > > >
> > > > 3. **Video results**. We have included a discussion at the very last paragraph of the paper to make clear that the positioning of our paper is not to compare with state of the art video generation models. We leave such a full exploration as future work.
> > > >
> > > > 4. Video samples are included in the supplementary material, please download the zip ball to view it.
> > > >
> > > > 5. We have completed the baseline plots in Figure 4.

---

### Official Review · Reviewer_Z34F · 2023-10-31

**Soundness:** 2 fair
**Presentation:** 3 good
**Contribution:** 3 good
**Rating:** 6
**Confidence:** 4

**Summary:**

This paper presents a multi-stage image/video generation model based on f-DM. The model is structured as a NestedUNet: a UNet with multiple inputs/outputs of increasing/decreasing resolutions. Compared to f-DM, it incorporates progressive growing and benchmarks the approach on multiple large-scale text-to-image and text-to-video datasets. The works positions itself as a new paradigm for high-resolution diffusion models and rivals latent DMs and cascaded DMs. It features faster training convergence in terms of the amount of iterations compared to the existing paradigms. The obtained results visually look quite good to me.

**Strengths:**

- Visually, the results look very good. And it is especially remarkable given the little compute used to train the models.
- A good advantage of the given method is that, compared to CDMs, it does not need require the previous stage to be well-trained to have meaningful training of the current stage with reliable scores. For CDMs, while it's possible to train all the stages simultaneously, one cannot generate images from scratch in the middle since the base stage has not been fully trained yet. Somehow, this does not happen for the given model.
- The ablations and evaluation in general is quite solid.
- The exposition is good, and the paper is written well.

**Weaknesses:**

- The method is not end-to-end (or at least does not perform well when trained in the end-to-end manner), despite what the paper claims. If I am not mistaken, at the end it still trains stage-by-stage similarly to CDMs — and without such stage-by-stage training it produces considerably worse results.
- The paper does not report training costs rigorously for all the experiments, and it's impossible to compare between methods without knowing their training cost.
- The comparison to CDMs does not seem fair, since the paper compares to under-trained CDMs. If one trains CDMs withing a limited computational budget, then more focus should be put on the base stage, since the final stages converge much faster.
- FID scores on ImageNet are ~3x times higher than the current SotA (e.g., MDT).
- The paper makes a claim about good results on a small text-to-image dataset (CC12M), but does not compare to existing large-scale text-to-image generators. This makes it impossible to evaluate this claim — e.g., Figure 8c should contain the results of existing text-to-image generators to make the existing model comparable. Otherwise, such a claim is not grounded. Judging by the maximum CLIPScore on a CLIP/FID trade-off chart is wrong since in the Imagen's paper, one can notice that even their bad models can attain very high CLIP scores under a strong enough guidance.

**Questions:**

- What are the computational budgets of all the experiments? (I can only see the amount of GPUs being used for the experiments — without a notice on for how long). I believe that the smaller amount of training could also justify inferior FID results on ImageNet compared to SotA.
- Are there any other differences compared to f-DM [1] apart from the progressive training idea and larger-scale experiments? It seems that f-DM uses the same NestedUNet idea, but the f-DM authors just do not call it a "NestedUNet". Do you use the same noise schedule as f-DM?
- To be honest, I do not quite understand why the FID on ImageNet is so high. Samples in Figure 5 looks very good to me (given that they are random samples). What CFG weight was used generate them?
- Please, include the comparison with existing text-to-image generators. Judging by the maximum attainable CLIP score is misleading.
- I find the results on video generation to be quite good. For how long has the model been trained and was there joint image/video training (or image pretraining) used?
- Why do you think your model does not suffer from the "train/test gap" problem of CDMs and its late stages can generate meaningful images even when the low-resolution stage has not been trained yet?
- I have a suspicion that the video generator can struggle in generating videos with moving scenes because the base low-resolution generator produces just a single frame. Could you please provide the video results for moving scenes? And in general include some mp4/gif videos in the submission (i have not found any in the supplementary).

Some typos:
- "crtical" => "criticial" (page 2)
- "under performs" => "underperforms" (page 6)
- "eg" => "e.g." (page 7)
- page 8 — no space before the bracket "(" in multiple places.

[1] Gu et al "f-DM: A Multi-stage Diffusion Model via Progressive Signal Transformation"

---

> ### Author Response · Authors · 2023-11-21
> **Response 1/N**
>
> We thank the reviewer for acknowledging our contributions and results, and for the comprehensive review and constructive questions. We answer your questions below, and we will provide more details with the updated draft.
>
>
> > The method is not end-to-end
>
> First, we apologize for misunderstandings on the end-to-end claim in our presentation. We agree that our progressive training mode does not exactly fit into the common perception of end-to-end training, and we are happy to remove such claims in the paper. However, we’d like to highlight that even without progressive training, MDM still offers significant advantage over simple diffusion, as well as our own implementations of CDMs and LDM (see Figure 4). In addition, the inference mode of MDM follows a single inference pass in the same latent space, with or without progressive training, which also differentiates itself from multi-stage approaches like CDM, LDM and fDM.
>
>
> > The paper does not report training costs rigorously for all the experiments, and it's impossible to compare between methods without knowing their training cost.
>
>
> We agree, thanks for pointing it out. Roughly speaking, the cost per iteration is near identical between single diffusion, MDM and CDM for the high resolution training, all are more costly that LDM. We will provide the actual training cost in the updated draft.
>
>
> > The comparison to CDMs does not seem fair, since the paper compares to under-trained CDMs. If one trains CDMs withing a limited computational budget, then more focus should be put on the base stage, since the final stages converge much faster.
>
> We agree that our own CDM baseline is not optimally tuned, however we do believe that it’s still a fair comparison and reflects the advantage of MDM over CDM. As correctly stated by the reviewer, CDM is sensitive to quality of the low res model, which requires one to both train the low resolution model to near optimal as well as carefully tune the noise augmentation level for the low resolution inputs. The comparison in Figure 4 shows that, starting from the same low resolution model, MDM clearly shows an advantage which demonstrates the robustness of our progressive training scheme over cascaded diffusion.
>
>
> > FID scores on ImageNet are ~3x times higher than the current SotA
>
>
> Our design for the ImageNet experiments are mostly focused on the control experiments, and we have directly borrowed the same architecture and hyper parameters from our text2image experiments. Also, due to computational demands, we did not train our models with a lot of iterations. This means that we have not optimized our FID results on ImageNet, and Table 1 is a rather a pessimistic estimate of our model’s performance. We are working on improving the results, and will update Table 1.
>
>
> > The paper makes a claim about good results on a small text-to-image dataset (CC12M), but does not compare to existing large-scale text-to-image generators
>
>
> We are working on updating the draft to include a comparison with other text-to-image baselines. However, we respectfully disagree that high CLIP score is not indicative of a model’s perceptual quality — in our experience, CLIP score correlates much better with the sample’s perceptual quality than FID. The reviewer refers to the Imagen paper’s results and indicate that bad models are capable of achieving high CLIP scores, however we could not find such evidence (eg, in Figure 4(a) ). Can you elaborate which Figure/Table supports this claim? (the closest thing we could find is Figure A.11, however these plots are from the same model size and the FID differences are also small).

---

> ### Author Response · Authors · 2023-11-21
> **Response 2/N**
>
> > computational budgets of all the experiments
>
>
> We will update the draft with the exact computational budgets for the experiments. But roughly speaking, our ImageNet experiments are all using 8xA100 GPUs, and Text2Image experiments are using 32xA100 GPUs. All experiments run between 1-2 weeks, which correspond to <500K training iterations.
>
>
> > Are there any other differences compared to f-DM
>
> MDM is related to f-DM in terms of parameter sharing across different resolutions, however with one fundamental difference. In f-DM, different resolutions are chained up sequentially, whereas the network handles one resolution at a time. This means that for f-DM the network architecture is identical to that of a standard diffusion model when denoising the highest resolution. In MDM, all resolutions are processed in parallel by the NestedUNet, both during training and inference. This sequential vs parallel difference is subtle, but makes a big difference wrt their empirical performance.
>
>
> > Figure 5 CFG weight
>
> Figure 5 is generated with  a CFG weight of 2. We have inspected results from different weights within the range [1.1, 5], they mostly visually appealing which indicates that CFG is applicable to MDM similarly to a standard diffusion model.
>
>
> > For how long has the (video) model been trained and was there joint image/video training (or image pretraining) used?
>
>
> The video model was trained following the same protocol as the image domain, whereas we first train an image generator for ~400K iterations. Then the video model is trained with the (temporal) NestedUnet, whereas there is a video loss and image loss (similar to image losses in different resolutions). The training time of the video model is less than two weeks on 32xA100 GPUs.
>
>
> > Why do you think your model does not suffer from the "train/test gap" problem of CDMs and its late stages can generate meaningful images even when the low-resolution stage has not been trained yet?
>
>
> The key difference between progressively training MDM and CDM is that for MDM, all parameters are jointly optimized, whereas CDM freezes the low resolution generator. In other words, the role of progressive training in MDM can be viewed as a better initialization of the weights. During inference time, MDM generates low and high resolution images in parallel, where we rely on a good inference schedule to deal with the training/test gap issue.
>
>
> > I have a suspicion that the video generator can struggle in generating videos with moving scenes because the base low-resolution generator produces just a single frame
>
>
> The reviewers hypothesis is correct and we will provide additional examples in the supplementary material.

---

> > ### Comment · Reviewer_Z34F · 2023-11-22
> > **Following up on the discussion**
> >
> > I am thankful to the authors for providing their response, it helped me to understand their work better.
> >
> > > About the end-to-end nature.
> > I believe this claim either should be reformulated with more caution, or more experiments and results should be provided with end-to-end training (e.g., most of the results should become about the end-to-end trained model).
> >
> > > About FID scores on ImageNet
> > Frankly, it is difficult to complain for me about FID on ImageNet (since, from my personal experience, one needs to evolve the entire project around ImageNet to get good FID on it), but such scores just make it difficult to position the method among the existing ones. How can I conclude from your exposition that it's just not enough training time instead of some inherent pathological property of the method that leads to high FID scores (e.g., what if the model stops improving below this FID)? Would it be possible to fine-tune MDM vs CDM from some base well-performing pixel-space diffusion (e.g., EDM)?
> >
> > > About FID/CLIPScore and comparison to existing text-to-image models.
> > After reading your response, I withdraw my claim that bad models can achieve high CLIPscore: I checked Imagen (and a couple of other papers) once again and couldn't find enough evidence to support it (the only plot that could support this is indeed A.11 in Imagen, but the majority of their plots show quite the opposite, i.e. that here you are right and I am wrong). I apologize for bringing this up. However, another part of my concern of reporting CLIPScores for existing generators still seems valid.
> >
> > > About MDM's vs CDM's train/test gap
> > Could you please provide the details about your "inference schedule to deal with the train/test gap" or point out to a paper's section where it is described? I checked Section 3.1 and "Implementation details" in Section 4.1 and couldn't find any details on it.
> >
> > Several my concerns have been resolved, and I've updated my score accordingly.
> > However, some of my concerns still remain:
> > - The claim about the end-to-end nature is not entirely justified
> > - Fig 4 feels confusing and does not really distinguish the convergence of different methods, since various methods use various pre-trained components (MDM/CDM/LDM), which can have different influence on the convergence of their subsequent stage. In this way, I cannot confidently arrive to the conclusion about the better convergence of MDMs in general (which is implied from the work). What feels more possible is that MDMs, within some fixed computational budget, would achieve better performance than CDMs/LDMs.
> > - Training costs (in terms of GPU days/years) are reported too vaguely and it's unclear whether the calculations include the pre-training stage or describe only the main training stage.
> > - The situation with too high FID scores on ImageNet is somewhat unclear to me (but at the same time I do not think that it's reasonable from my side as a reviewer to require the authors to tune their method exclusively on ImageNet). It's unclear to me what conclusion should I make from the provided experiments on ImageNet.
> > - Lack of video results
> > - Positioning the developed text-to-image/text-to-video model in terms of the scores within the existing methods
> > I am looking forward to the manuscript update to further revise my score.

---

> > > ### Author Response · Authors · 2023-11-23
> > > **Draft Updated**
> > >
> > > We thank the reviewer for the prompt response and additional feedbacks. We have updated the draft which we believe should address most of your remaining concerns.
> > >
> > > 1. **end-to-end claim**. We agree with the reviewer that this causes confusion, and have removed all end-to-end related claims.
> > > 2. **ImageNet results**. We have updated Table 1 with much improved FID results on ImageNet. As stated before, we did not optimize our hyper parameters specifically for ImageNet, and also was not training the models long enough. During the rebuttal period, we have tried to increase the training batch size, and also fixed our previous way of performing CFG (we were not properly dropping labels during training in the previous version). After these simple changes, we see drastic improves in the FID numbers which now match the baseline results in Table 1. We are optimistic that there is yet more room to improve with some more hyper parameter tuning. (BTW, we really appreciate your understandings on the point about optimizing the project around ImageNet.)
> > >
> > > 3. **CLIP vs FID**. Thank you for acknowledging our points and we are very glad that we could reach an agreement on this subtle matter :) We also agree with the reviewers suggestion and have updated Figure 8 to include the results from the Imagen paper as well. We have also edited the discussions in Sec 4.3 to reflect this update.
> > > 4. **Inference details**. We have provided such details in Figure 11 and Sec C in the updated draft. The simplest way to understand the inference schedule is to think of MDM as performing standard diffusion in an extended space, where the space is a concatenation of different resolutions. This enables a smooth generation process across all resolutions in parallel, and we have no more train/test mismatch than that of a standard diffusion model (in principle). CDM on the other hand adopts a stage wise inference procedure, whereas the train/test gap can be easily prominent between two stages.
> > > 5. **Fig 4 and convergence speed**. First of all, we agree with the reviewer's assessment. It is indeed very difficult to conclude that MDM converges faster than CDM or LDM as we can not control the pretraining stage of CDM or LDM perfectly. As a result, we have been careful not to claim that MDM is superior to CDM/LDM wrt convergence. However, the comparison between MDM and simple diffusion is straightforward and fair, and we have clear evidence that MDM offers a much better solution than simple diffusion in high resolution space. Our view of MDM is thus an alternative that can be a favorable choice over other designs in certain scenarios, but should always be preferred over simple diffusion.
> > > 6. **Total training costs**. We apologize for the lack of clarity on this matter, and have updated the draft with detailed documentation of the training costs of our main models (see Table 3).
> > > 7. We have provided video samples in the supplementary material (please download the zip ball). (btw there are also a lot more visualizations in the appendix).
> > > 8. **Positioning**. We have added discussions at the end of the paper (the very last paragraph) to clarify the positioning of our work.
> > >
> > > And lastly, we really enjoyed your reviews and thanks again for the thoughtful feedbacks and discussions :)

---

### Official Review · Reviewer_iXvr · 2023-11-01

**Soundness:** 3 good
**Presentation:** 3 good
**Contribution:** 3 good
**Rating:** 8
**Confidence:** 4

**Summary:**

This paper introduces a new framework for high-resolution image synthesis using diffusion models. Due to computational limitations, diffusion models are often limited to cascaded approaches in pixel-space or operating in latent space. The proposed framework, Matryoshka Diffusion Models (MDMs), denoises images at various resolutions simultaneously. MDM is trained using the standard diffusion objective jointly at multiple resolutions with a progressive schedule where the higher resolutions are added into the objective later in training. The authors demonstrate that MDM has greater efficiency with comparable performance on image and video generation.

**Strengths:**

- The proposed framework is straightforward and easy to understand, drawing inspiration from existing GAN literature to address limitations of current diffusion models for high-resolution synthesis.
- The authors evaluate MDM on multiple synthesis tasks and demonstrate comparable results. The qualitative results look impressive especially given the relatively small scale of training data.
- Ablation studies quantify the effects of progressive training and the number of nested levels on the quality and alignment of the outputs.

**Weaknesses:**

- The variables in the provided pseudocode for the NestedUNet architecture are not clear. A quick description to clarify the inputs to the function would be beneficial.
- In Table 1 we see that there is a noticeable gap between MDM and the baselines. It would be helpful if the authors provided insight as to why they think this may be the case.
- The authors highlight video generation as a contribution of MDM, but there is no discussion of the experiments or results (aside from implementation details and a few subsampled frames). It is difficult to get a sense of MDM's performance for this task.
- There are several typos (especially with spacing before citation parentheses on page 8).

**Questions:**

- Generally the experiments section would benefit from including more insights on the results, especially for the scenarios where MDM is outperformed by the baselines.
- While there is some differentiation already denoted through the colors, it would be helpful to explicitly label the novel pathways introduced by the NestedUNet architecture in Figure 3 to clearly distinguish from skip connections from the original UNet.
- It is helpful to understand how the proposed multi-resolution prediction affects sampling speed.

---

> ### Author Response · Authors · 2023-11-21
>
> We’d like to thank the reviewer for acknowledging our contributions and results. We answer your questions below, and we will provide more details with the updated draft.
>
>
> > The variables in the provided pseudocode for the NestedUNet architecture are not clear
>
>
> We will update to pseudocode to clarify this.
>
>
> > In Table 1 we see that there is a noticeable gap between MDM and the baselines.
>
>
> Our design for the ImageNet experiments are mostly focused on the control experiments, and we have directly borrowed the same architecture and hyper parameters from our text2image experiments. Also, due to computational demands, we did not train our models with a lot of iterations. This means that we have not optimized our FID results on ImageNet, and Table 1 is a rather a pessimistic estimate of our model’s performance. We are working on improving the results, and will update Table 1.
>
>
> > Video results
>
> Due to resource constraints, the experimental results on the text-to-video tasks are preliminary, and our goal is to demonstrate qualitatively that the same idea behind MDM also applies to video generation. There is great interest to us to further improve the video generation’s quality and compare it with state of the art methods, which we leave as future work.
>
>
> > Sampling speed
>
> MDM’s sampling speed is comparable to that of a standard UNet, and we will provide measurements in the updated draft.

---

> > ### Comment · Reviewer_iXvr · 2023-11-22
> >
> > Thank you to the authors for addressing the reviewers' concerns. While I still believe this work is very interesting, the other reviewers have pointed out some incompleteness in the evaluations that I initially missed and think are valuable points to more concretely address than in the initial replies by the authors in order to maintain a positive score. I am interested in seeing these concerns addressed in the revised draft before updating my final score.

---

> > > ### Author Response · Authors · 2023-11-23
> > > **Draft updated**
> > >
> > > We thank the reviewer for the prompt responses. We have updated the draft with more results and details which we believe answers your questions.
> > >
> > > 1. We updated the pseudo code, as well as the illustration Figure 3 for better clarity. We also provided more details on the inference procedure in Figure 11 and Sec C in Appendix.
> > >
> > > 2. We have updated Table 1 with much improved FID results on ImageNet. As stated before, we did not optimize our hyper parameters specifically for ImageNet, and also was not training the models long enough. We have tried to increase the training batch size, and also fixed our previous way of performing CFG (we were not properly dropping labels in the previous version). After these simple changes, we see drastic improves in the FID numbers which now match the baseline results in Table 1. We are optimistic that there are yet more room to improve with some more hyper parameter tuning.
> > >
> > >  3. We have provided video samples in the supplementary material; we have also added discussions (the last paragraph of the paper) to make clear that we are not claiming that our video results are comparable to the SOTA, but rather to leave such an exploration as future work.
> > >
> > > 4. We have provided a benchmarking result of Nested UNet vs standard UNet wrt their inference speed in Table 2. This shows that Nested UNet (and hence MDM) has the same inference speed as a standard diffusion model with UNet.
> > >
> > > 5. Concerns raised by other reviewers are also addressed in this update, please refer to the specific sections in the draft, and/or our responses to the specific questions under other reviews.

---

### Official Review · Reviewer_7XqF · 2023-11-01

**Soundness:** 2 fair
**Presentation:** 3 good
**Contribution:** 3 good
**Rating:** 6
**Confidence:** 4

**Summary:**

The paper proposes a diffusion model capable of denoising multiple resolutions simultaneously. To enhance computational efficiency, they use a nested UNet structure. While it is possible to train the model all at once, the results show that progressively training it led to better convergence.

**Strengths:**

The proposed model converges faster compared to traditional cascaded diffusion models. In the case of MS-COCO, the model demonstrates superior performance.

**Weaknesses:**

It is anticipated that there will be an increase in computational load. And the performance does not seem to surpass that of LDM, which employs classifier-free guidance. Also, since multi-ratio and resolution training is already being conducted in stable-diffusion XL, the proposed method with multi-resolution training is not much novel.

**Questions:**

It is curious about the computational load increases. Also, it would be nice if the performance difference with usual UNet and NestedUNet is provided.

---

> ### Author Response · Authors · 2023-11-21
>
> We thank the reviewer for the constructive feedbacks. We answer some of your questions below, and we will provide more details with the updated draft.
>
>
> > It is anticipated that there will be an increase in computational load
>
>
> Note that the computational cost of Nested Unet is almost identical to a standard Unet, as shown in Figure 3.
>
>
> > And the performance does not seem to surpass that of LDM, which employs classifier-free guidance
>
> In Figure 4, we show the comparison of MDM to our own implementation of LDM. The LDM baseline uses the same Unet architecture as our 64x64 Unet, and we see that MDM outperforms LDM in this controlled setting. Note also that we didn’t apply CFG to either MDM or LDM, and we show in Figure 8 that CFG is applicable to MDM, similar to that shown in LDM.
>
>
> > Also, since multi-ratio and resolution training is already being conducted in stable-diffusion XL, the proposed method with multi-resolution training is not much novel.
>
> We agree that multi resolution training in SDXL is related to MDM, but we respectfully point out that they work in fundamentally different ways. In SDXL, multi-resolution training can be considered as form a scheduled data augmentation, where the same model is applied to data of different resolutions, and during inference, the model is only capable of generating images of the highest resolution. MDM on the other hand, consumes images of multiple resolutions as part of the model architecture as well as the inference pipeline. It allows one to train images of multiple resolutions at the same time, and also generate images of multiple resolutions in a single inference run (see Figure 1 (a) top left corner as an example).
>
>
> > Also, it would be nice if the performance difference with usual UNet and NestedUNet is provided
>
> We will provide additional experiments training a single NestedUnet and standard Unet in the updated draft — our observation is that they have near identical performance as two stand alone architectures.

---

> > ### Author Response · Authors · 2023-11-23
> > **Draft updated**
> >
> > We have provided in the updated draft with comparison between Nested UNet and a standard UNet, both wrt training performance, and training/inference speed. Please refer to Figure 12 and Table 2 in the appendix for details. As stated above, these two architectures have almost identical training and inference performance and costs. Which means that the performance gains from MDM are primarily from the multi resolution diffusion process.

---

> > > ### Comment · Reviewer_7XqF · 2023-12-03
> > >
> > > Thank you for clarifying my comments. I misunderstanded the network architecture and now understand that Nested UNet has similar computation costs with original UNet. Also, although adding multi-resolution losses was a common idea in GAN literature, I agree there is a lack of study to use multi-resolution loss for diffusion framework. I updated my score above.

---

### Author Response · Authors · 2023-11-23
**Updated Draft**

We thank all reviewers for their constructive feedbacks. We have uploaded a significantly improved draft, together with a separate supplementary material file. Here are the import edits/changes to the draft.


1. We have provided additional experimental results, including.
    1. Much improved FID results on ImageNet. This is achieved by training with a larger batch size, and properly performing CFG. Our best FID now is improved to 3.51 compared to the originally reported 6.62.
    2. Updated Figure 4 to complete the baseline plots which we were unable to obtain in the previous version.
    3. A lot more qualitative results in the appendix, and also provided additional video samples in the supplementary material (please download the zip ball to view)
    4. Comparisons between UNet and Nested UNet, which is shown in Figure 12 and Table 2 in the appendix.
    5. Exact report of training cost for our main experiments in Table 3.
    6. Training speed measurements for baseline models in Table 4.
2. Inference procedure is elaborated in Sec C in appendix, and illustrated in Figure 11.
3. We have added more implementation details in the Appendix
4. More discussions on the results (Sec 4.2), as well as discussion on the positioning of our method in relation to SOTA text2image/text2video models (end of Sec 6)
5. We have made edits to the claims and positioning of the paper. We dropped claims related to the end-to-end training argument to avoid confusion.
6. We updated the illustration in Figure 3 to better contrast it with UNet.
7. We fixed all typos and formatting issues as pointed out by the reviewers.

---

### Meta-Review · Area_Chair_XQWi · 2023-12-09

**Metareview:**

This paper presents an alternative to latent diffusion models or cascaded models for learning diffusion models on high-resolution images, by training a model jointly across different resolutions. The reviewers have acknowledged the extensivity of the experiments and the quality of writing, however, they originally raised concerns regarding missing details such as the run time and video results. In the rebuttal, the authors improved the submission by providing additional info. The ACs are happy to recommend acceptance following the reviewers' recommendation.

**Justification For Why Not Higher Score:**

The proposed ideas are not ground-breaking, but are important and of interest to the community.

**Justification For Why Not Lower Score:**

The reviewers support accepting the submission. Most major concerns were addressed in the rebuttal.

---

### Decision · Program_Chairs · 2024-01-16

Accept (poster)